# scSSL-Bench: Benchmarking Self-Supervised Learning for Single-Cell Data

Olga Ovcharenko [* 1]   Florian Barkmann [* 2]   Philip Toma [* 2]   Imant Daunhawer [2]   Julia E. Vogt [2]
Sebastian Schelter [† 1]   Valentina Boeva [† 2 3 4]

## Abstract

Self-supervised learning (SSL) has proven to be a powerful approach for extracting biologically meaningful representations from single-cell data. To advance our understanding of SSL methods applied to single-cell data, we present scSSL-Bench, a comprehensive benchmark that evaluates nineteen SSL methods. Our evaluation spans nine datasets and focuses on three common downstream tasks: batch correction, cell type annotation, and missing modality prediction. Furthermore, we systematically assess various data augmentation strategies. Our analysis reveals task-specific trade-offs: the specialized single-cell frameworks, scVI, CLAIRE, and the finetuned scGPT excel at uni-modal batch correction, while generic SSL methods, such as VICReg and Sim-CLR, demonstrate superior performance in cell typing and multi-modal data integration. Random masking emerges as the most effective augmentation technique across all tasks, surpassing domain-specific augmentations. Notably, our results indicate the need for a specialized single-cell multi-modal data integration framework. scSSL-Bench provides a standardized evaluation platform and concrete recommendations for applying SSL to single-cell analysis, advancing the convergence of deep learning and single-cell genomics.

## 1. Introduction

Recent progress in single-cell RNA sequencing (scRNA-seq) and multi-omics sequencing technologies has transformed our understanding of cellular heterogeneity by enabling cell molecular profiling at unprecedented resolution (Sikkema et al., 2023; Eraslan et al., 2022). This breakthrough has revolutionized our ability to understand diseases, develop personalized treatments, and trace the origins of complex conditions like cancer and autoimmune disorders. scRNA-seq (Tang et al., 2009) captures gene expression levels in individual cells and generates a high-dimensional matrix where each row represents a cell and each column represents a gene's expression level. Multi-omics approaches simultaneously measure additional molecular features, including chromatin accessibility through ATAC-seq (Grandi et al., 2022) or protein levels via CITE-seq (Stoeckius et al., 2017). Modern multi-omics experiments generate massive datasets encompassing hundreds of thousands of cells, with each cell characterized by diverse measurements: the expression of tens of thousands of genes, the accessibility of hundreds of thousands of chromatin regions, and the abundance of hundreds of surface proteins. Multi-modal profiling provides an unprecedented view of cellular state and function. However, the resulting datasets are susceptible to *batch effects* — systematic technical variations introduced during sample preparation, sequencing, or processing (Lähnemann et al., 2020). If left uncorrected, batch effects mask genuine biological signals and compromise downstream analyses (Heumos et al., 2023). For instance, when comparing blood samples from cancer patients processed in different laboratories, batch effects can make immune cells from the same patient appear more different from each other than from cells of other patients, masking crucial patterns in how the immune system responds to the tumor (Slyper et al., 2020).

The success of self-supervised learning (SSL) methods in computer vision (He et al., 2020; Chen et al., 2020), video processing (Schiappa et al., 2023), and natural language processing (Min et al., 2023) has inspired their application to single-cell data. Several models have been adapted for analyzing single-cell data (Han et al., 2022; Liu et al., 2024; Li et al., 2023; Tang et al., 2024), showing promising results in mitigating batch effects and improving downstream analyses. There is an interest in the genomics community in finding standardized approaches for applying SSL to single-cell analysis. A recent work (Richter et al., 2024a) discusses scenarios in which SSL is applicable to single-cell genomics.

---

[*†] Equal contribution [1]BIFOLD & TU Berlin, Berlin, Germany [2]Department of Computer Science, ETH Zurich, Zurich, Switzerland [3]Swiss Institute of Bioinformatics, Lausanne, Switzerland [4]Paris Cité University, Cochin Institute, INSERM U1016, Paris, France. Correspondence to: Olga Ovcharenko <ovcharenko@tu-berlin.de>, Sebastian Schelter <schelter@tu-berlin.de>, Valentina Boeva <valentina.boeva@inf.ethz.ch>.

*Proceedings of the 42$^{nd}$ International Conference on Machine Learning*, Vancouver, Canada. PMLR 267, 2025. Copyright 2025 by the author(s).

The authors compare the performance of masked autoencoders and two SSL methods (BYOL (Grill et al., 2020) and Barlow Twins (Zbontar et al., 2021)), discuss the effects of pre-training on auxiliary data, and empirically study the efficacy of zero-shot and fine-tuned SSL. However, their work lacks a comparison to specialized single-cell models and does not explore individual hyperparameters and regularization techniques. Furthermore, building upon innovations in natural language processing (Vaswani et al., 2017; Devlin et al., 2019b), single-cell foundation models (Cui et al., 2024; Yang et al., 2022a; Theodoris et al., 2023) have recently emerged as powerful tools to understand cellular heterogeneity and gene-gene interactions, and require a comparison to contrastive methods.

Since the majority of SSL methods were originally developed for image and text data, there is a lack of a systematic comparison of models, hyperparameters, training regimes, regularization techniques, and augmentations for single-cell genomics data (Toma et al., 2024). This knowledge gap limits our understanding of how to effectively adapt and optimize SSL methods for single-cell data. Our work seeks to fill this gap and focuses on the following research questions:

- *RQ1* – Do specialized single-cell SSL methods outperform generic SSL methods? How does the performance of SSL models differ for uni-omics and multi-omics data?

- *RQ2* – How do hyperparameters and augmentation techniques impact the performance of generic SSL methods for single-cell data?

- *RQ3* – Are batch normalization and multi-modal integration techniques proposed for image data beneficial for single-cell genomics data as well?

Our main contribution is an open-source benchmark, scSSL-Bench, which compares the performance of several self-supervised learning methods for single-cell data. (1) To address *RQ1*, we evaluate nineteen generic and specialized single-cell SSL methods across seven different single-cell uni-modal and two multi-modal datasets, assessing their performance on three common downstream tasks: batch correction, cell type annotation, and missing modality prediction (Subsection 4.1). Our results reveal that specialized frameworks, scVI and CLAIRE, together with the foundation model, scGPT, are the best for uni-modal batch correction, while generic SSL techniques such as VICReg and SimCLR outperform domain-specific methods for multi-modal batch correction and the other two tasks on single-modal data. (2) For *RQ2*, we evaluate various model architectures and hyperparameters, including representation and projection dimensionality, augmentation strategies, and multi-modal integration methods. (Subsection 4.2). Overall, we find that a moderate to larger embedding dimensionality consistently leads to improved results and identify masking as

the most beneficial augmentation technique that surpasses biology-specific augmentations. (3) An assessment of design decisions suggested in the related work, e.g., retaining projector and domain-specific batch normalization, helps to answer *RQ3* and to find best practices that can be adopted by the single-cell genomics community (Subsection 4.3). We find that neither domain-specific batch normalization nor retaining the projector during inference improves results.

We provide our benchmark code under an open license at `https://github.com/BoevaLab/scSSL-Bench` for reproducibility and for fostering further research on SSL for single-cell data.

## 2. Background

In this Section, we discuss SSL on single-cell data, the corresponding downstream tasks, and specialized methods.

### 2.1. Machine Learning on Single-Cell Data

**Data:** There exist multiple technologies that measure different aspects of the cellular state. scRNA-seq (Tang et al., 2009) measures which genes are expressed in each cell and produces a high-dimensional sparse count matrix representing individual cells as rows and genes as columns. CITE-seq (Stoeckius et al., 2017), 10x multiome (Baysoy et al., 2023), and TEA-seq (Swanson et al., 2020) profile complementary to gene expression aspects such as chromatin accessibility and protein abundance within a cell.

**Downstream Tasks:** Learned cell representations are commonly used for multiple downstream tasks.

*Batch correction* – Single-cell data can be affected by batch effects, which challenges the ability to measure true biological variation (Yu et al., 2023; Polański et al., 2019). Batch effects are technical biases introduced while sequencing because of differences in sequencing platforms, timing, reagents, or experimental conditions across laboratories (Zhang et al., 2024). To address the presence of batch effects, a common approach is learning a batch-corrected lower-dimensional embedding, where cells cluster based on their cell type and cell state rather than their experimental batch of origin (Hao et al., 2024) (Figure G2 in the Appendix illustrates cells before and after batch correction).

*Cell type annotation* – This task (also called as query-to-reference mapping) revolves around unsupervised transfer learning (Yang et al., 2022b), where the primary objective is to annotate cells of a hold-out dataset (query) by mapping them to a joint latent space of a pre-annotated train dataset (reference), whose cell types are known (Lotfollahi et al., 2022). Once test and train data are aligned, held-out cells are annotated using a classifier trained on embeddings of the reference dataset. Figure G3 in the Appendix visualizes how

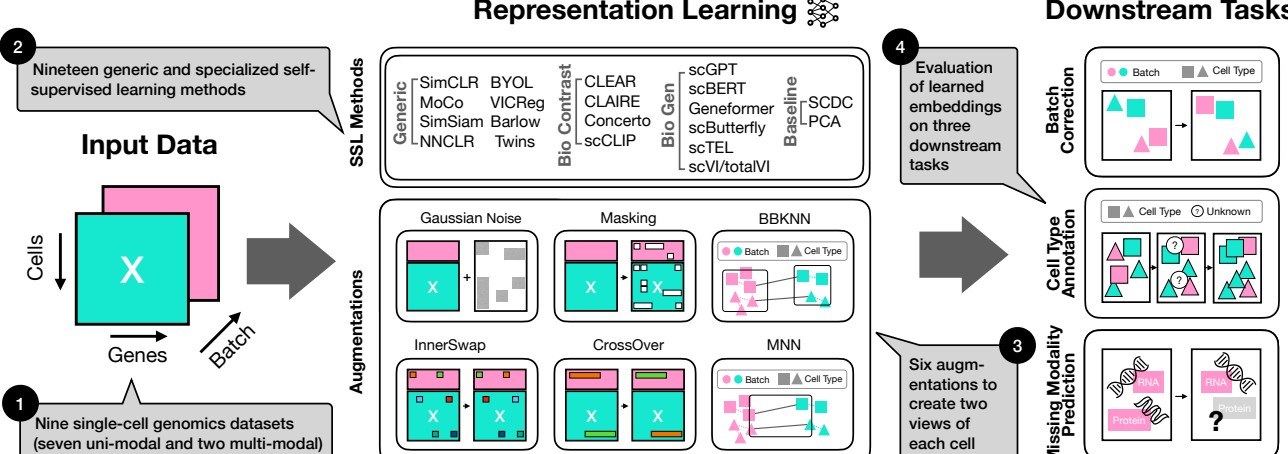

*Figure 1.* Outline of scSSL-Bench: ❶ As input, scSSL-Bench takes scRNA-seq data (cell-by-gene count matrix), where each value in the matrix represents the number of reads in a cell for the corresponding gene. ❷ scSSL-Bench trains one of nineteen methods: Generic, specialized contrastive (Bio Contrast), specialized generative (Bio Gen), and baselines. For self-supervised generic methods, scSSL-Bench uses augmentations ❸ to create two views of a cell. ❹ The learned embeddings are evaluated on three downstream tasks.

the learned representations of train and hold-out sets and train cell types are used to predict the cell types of hold-out data (query, blue result) during subsequent inference.

*Missing modality prediction* – For multi-modal datasets, missing modality prediction enables the inference of unmeasured (missing) modalities in held-out (query) cells (Yang et al., 2022b). Given multi-modal train data (reference) with RNA and protein expressions and hold-out data containing only RNA, the goal is to predicts the hold-out dataset's original protein values by averaging the proteins of the nearest neighbors from the train set (referred to as kNN probing).

### 2.2. Self-Supervised Learning (SSL) Methods

SSL aims to discover useful data representations without relying on annotations (Geiping et al., 2023) by leveraging the dis-/similarity of data samples. We refer to Appendix A for details on generic SSL methods (and their contrastive and non-contrastive variants).

**Single-Cell Contrastive Methods:** There are several SSL frameworks tailored for single-cell data. CLEAR (Han et al., 2022) employs contrastive SSL and leverages InfoNCE loss (van den Oord et al., 2019). Positive/negative pairs are created by adding Gaussian noise, random masking, or crossing over genes between two cells. CLAIRE (Yan et al., 2023) suggests a novel augmentation strategy by finding mutual nearest neighbors (MNN) between and nearest neighbors (KNN) within experimental batches in a dataset. CLAIRE uses inter-biological-batch MNN pairs as initial positive pair seeds, which are then "mixed" with intra-biological-batch neighbors to generate positive pairs. CLAIRE extends MoCo's (He et al., 2020) architecture with online and momentum encoders. Concerto (Yang et al., 2022b)

is a contrastive self-supervised distillation framework that uses an asymmetric teacher-student network structure (Hu et al., 2023) and dropout to create two augmented views of a cell (a positive pair). Positive/negative pairs are contrasted using NTXent loss (Sohn, 2016). Concerto also supports single-cell multi-modal data, e.g., pairs of RNA and protein. The scCLIP (Xiong et al., 2023) method is a generalized multi-modal transformer model that applies contrastive learning to single-cell multi-omics data, which adopts ideas from CLIP (Radford et al., 2021) by defining modality-specific encoders, constructing positive pairs from two modalities of the same cell, and contrasting them using InfoNCE loss (van den Oord et al., 2019).

**Single-Cell Generative Methods:** Generative approaches, from variational autoencoders (VAE) to transformer-based foundation models, are used to learn single-cell representations and exploit the biological batch/cell type annotations during training, which can be seen as leaking information compared to contrastive SSL methods. The state-of-the-art single-cell method scVI (Lopez et al., 2018) is a widely-used VAE that leverages a zero-inflated negative binomial distribution as reconstruction loss. A multi-modal version of scVI, totalVI (Gayoso et al., 2021), allows joint analysis of RNA and protein expressions. SCDC (Li et al., 2024) is a uni-modal method that employs biological batch and cell type encoders to create a concatenated representation that is reconstructed using the decoder. To improve the discrimination of the batch information, SCDC uses a specialized batch discriminator. scTEL (Chen et al., 2025) leverages transformer and LSTM layers to establish a mapping from RNA expression to unobserved protein expression in the same cells. scButterfly (Cao et al., 2024b) supports CITE-seq (Stoeckius et al., 2017) data and employs a dual-VAE architecture with modality-specific pretraining and a dis-

criminator to encourage the mixing of different modalities. Recently, transformer-based single-cell foundation models (scFMs) have emerged. For generalizability, scFMs are pre-trained on tens of millions of cells. scBERT (Yang et al., 2022a) adapts BERT's (Devlin et al., 2019a) masked language modeling to learn contextual gene representations, scGPT (Cui et al., 2024) uses GPT-style pretraining to create transferable representations across cell types and experimental conditions, and Geneformer (Theodoris et al., 2023) employs a transformer architecture pre-trained on large-scale gene expression datasets to capture gene-gene interactions.

## 3. Benchmark Design

Figure 1 illustrates our design of scSSL-Bench. The input of the benchmark are cell-by-gene count matrices containing scRNA-seq or CITE-seq data ❶. Depending on the data and SSL method, scSSL-Bench trains one of nineteen representation learning frameworks ❷ using augmentations to create positive/negative pairs ❸ for self-supervised approaches. Finally, the learned representations are evaluated on three downstream tasks ❹.

### 3.1. Datasets, Models, and Tasks

**Datasets:** We consider nine single-cell genomics datasets that represent common established benchmarks (Richter et al., 2024a). Peripheral Blood Mononuclear Cells (PBMC), Pancreas, Immune Cell Atlas, Mouse Cell Atlas (MCA), Human Immune Cells (HIC), Lung, and Tabula Sapiens are seven single-modal datasets collected using scRNA-seq (Tang et al., 2009) technology. Multi-modal Peripheral Blood Mononuclear Cells (PBMC-M) and Multi-modal Bone Marrow Mononuclear Cells (BMMC) are multi-modal datasets collected using CITE-seq technology that contain RNA and protein or gene expression and protein abundance (ADT) respectively. Further details in Appendix B.

**SSL Methods:** To investigate *RQ1*, we benchmark nineteen existing SSL methods and divide them into four categories: generic, domain-specific specialized contrastive and generative methods, and baselines. SimCLR (Chen et al., 2020), MoCo (He et al., 2020), SimSiam (Chen & He, 2020), NNCLR (Dwibedi et al., 2021), BYOL (Grill et al., 2020), VICReg (Bardes et al., 2022), and BarlowTwins (Zbontar et al., 2021) are generic SSL architectures that we adopt to single-cell (multi-omics) data (see Figure G1 for architecture details). Contrastive domain-specific methods that are tailored for the single-cell data include Concerto (Yang et al., 2022b), CLEAR (Han et al., 2022), CLAIRE (Yan et al., 2023), and scCLIP (Xiong et al., 2023). Generative methods include commonly used for single-cell data integration, scVI (Lopez et al., 2018) and totalVI (Gayoso et al., 2021), which are single-cell specialized variational autoencoder-based methods, and single-cell foundation

models scGPT (Cui et al., 2024), Geneformer (Theodoris et al., 2023), and scBERT (Yang et al., 2022a). Additionally, for multi-omics integration, we evaluate scButterfly (Cao et al., 2024b) and scTEL (Chen et al., 2025), which leverage variational autoencoders. SCDC (Li et al., 2024) and principal component analysis (PCA) (Pearson, 1901) are used as baselines in scSSL-Bench. We include PCA as a baseline to assess whether more complex SSL methods offer substantial improvements over a simple linear dimensionality reduction technique that does not correct for batch effects. We refer to Subsection 2.2 for detailed descriptions of each method.

First, in all contrastive methods except Concerto, two views are created by augmenting a single sample. Second, both views are encoded by a network with shared weights, producing data representations. Concerto removes the necessity for transforming samples by placing a dropout layer behind the encoder backbone. Finally, while training, all representations produced by the encoder are passed into a projector to improve robustness (Xue et al., 2024). In all contrastive approaches but Concerto and scCLIP, the projector is discarded during inference, keeping only the encoder's output.

**Downstream Tasks and Evaluation:** To address *RQ1*, our benchmark evaluates multiple single-cell datasets on three tasks: batch correction, cell type annotation, and modality prediction (see Subsection 2.1 for details).

*Batch correction* – the quality of batch-corrected embeddings is measured by biological conservation and batch correction metrics. These metrics were introduced in single-cell integration benchmarking (scIB) (Büttner et al., 2019; Luecken et al., 2022; Tran et al., 2020), a tool that is widely used in the single-cell community, see Appendix C for details. Analogous to Luecken et al. 2022, we combine bio conservation (Bio), measuring the similarity between cell embeddings and ground-truth cell types or states, and batch correction (Batch), measuring how well the batch effect is removed, by aggregating these scores into a total score by $Total = 0.6 \times Bio + 0.4 \times Batch$. All tables showing batch correction results are min-max scaled inside each dataset.

*Cell type annotation* – each dataset is divided into train (reference) and test (query) data that consists of up to three held-out (experimental) batches with unseen cells (details in Appendix C). We train a k-nearest neighbors (KNN) classifier with train (reference) embeddings and cell types to annotate test (query) data representation. Next, k-nearest neighbor probing (Marks et al., 2025) is used to predict cell types, and performance is evaluated using the macro-average F1-score and classification accuracy (Heryanto et al., 2024).

*Missing modality prediction* on multi-modal datasets – we evaluate the quality of the inferred modality by measuring the Pearson correlation between the original and predicted values, see Appendix C for more details.

## 3.2. Augmentation, Batch Normalization, and Multi-Modal Integration

**Augmentations:** We evaluate augmentations for single-cell data proposed in CLEAR (Han et al., 2022) and CLAIRE (Yan et al., 2023) to investigate *RQ2*. The purpose of augmentations in contrastive SSL is to transform the original sample into two distinct views that are contrasted during training (Zhang & Ma, 2022). Multiple augmentations can be applied to a data sample to improve the generalization and robustness of representations. The authors of CLEAR (Han et al., 2022) introduce four augmentations for scRNA-seq data, each of which we apply with 50% probability: Masking, Gaussian noise, InnerSwap, and CrossOver. First, a random mask sets 20% of a cell's genes to zero, followed by additive Gaussian noise (with mean 0 and standard deviation 0.2) to 80% of genes in the cell. Then, 10% of genes are swapped within the cell (InnerSwap), before mutating 25% of gene expression values with another random cell (CrossOver). CLAIRE uses a neighborhood-based approach: mutual nearest neighbors (MNN) in the unintegrated space are computed for each cell across all batches. During augmentation, an inter- and an intra-batch views are computed by mutating between neighboring cells (Yan et al., 2023). We also evaluate sampling positive pairs from a batch-balanced KNN (BBKNN) graph. We investigate the impact of the MNN and BBKNN augmentations on the batch correction performance.

**Domain-Specific Batch Normalization (DSBN):** Concerto (Yang et al., 2022b) adapts the idea of DSBN (Chang et al., 2019), a technique originally suggested for image data. DSBN helps to learn domain-specific information to produce domain-invariant representations by applying separate batch normalization layers for each domain. To investigate *RQ3*, we replace the common batch normalization with DSBN where each experimental batch (different laboratory experiment in the same dataset) gets its own batch normalization layer, similar to Concerto.

**Multi-Modal Integration:** For the multi-omics datasets PBMC-M and BMMC, we evaluate three integration methods as part of *RQ3*. First, addition takes two embeddings of the same dimensionality (one per modality) and adds them together to get a joint representation, similar to the Concerto (Yang et al., 2022b). Second, concatenation appends two embeddings. Third, instead of contrasting joint views of a cell, two modalities of the same cell are contrasted using a symmetric cross-entropy loss (Wang et al., 2019) and the CLIP approach. After training with the CLIP approach (Radford et al., 2021; Xiong et al., 2023), we concatenate two embeddings during inference.

## 4. Experiments

As detailed in Section 3, we benchmark nineteen SSL methods on nine single-cell datasets derived from different tissues with considerable variation in data size and complexity. See Appendix B for details about the datasets. All models are trained with five unique random seeds and we report their mean performance and standard deviation.

### 4.1. Generic versus Specialized SSL Methods

*RQ1* focuses on the comparison of specialized single-cell SSL frameworks and generic SSL methods. For that, we evaluate several models and two baselines on three important downstream tasks for uni- and multi-omics datasets.

**Batch Correction:** The batch correction performance of all methods across five datasets is presented in Table 1. Our analysis includes two multi-modal (CITE-seq) datasets (PBMC-M and BMMC) and three single-modality (scRNA-seq) datasets (PBMC, Pancreas, and Immune Cell Atlas).

For scRNA-seq datasets, our results show that scVI is the best-performing method that balances both batch correction and bio conservation (Table 1). scVI performance drops for the MCA and Lung datasets (Table H1) but excels for the Tabula Sapiens dataset (Table H11). CLAIRE ranks second-best overall but tends to overcorrect batch effects at the expense of biological variance. Overall and comparing to other single-cell generative models, finetuned scGPT performs well for the second-largest evaluated dataset, Immune Cell Atlas, by scoring high in bio conservation and total, but for smaller datasets batch score is significantly lower than other methods. Zero-shot scGPT and finetuned Geneformer show unsatisfactory performance. Across all common benchmarked SSL methods, VICReg, SimCLR, and MoCo perform satisfactorily for the Pancreas dataset. However, for the PBMC and Immune Cell Atlas datasets, these SSL methods prioritize batch correction over bio conservation as indicated by their high batch and low bio score. In comparison to the other methods, in all cases, Concerto significantly underperforms and achieves a total score lower than the baselines, PCA and SCDC. As expected, PCA shows an adequate bio conservation score since it uses raw data and captures the true biological signal.

For multi-modal datasets, PBMC-M and BMMC, we observe that generic methods such as SimCLR, BYOL, MoCo, and VICReg are the best-performing methods, within their category and overall (Table 1). Interestingly, MoCo overcorrects for all single- and multi-modal datasets. The results show that there is room for improvement in specialized methods. Concerto, scCLIP, and scButterfly reach low batch correction results compared to general methods as MoCo or SimCLR. For PBMC-M, Concerto preserves biological variance and shows a high bio score. scTEL succeeds at

*Table 1.* Batch integration performance across five datasets. We show each method's biological conservation score (Bio), batch correction score (Batch), and total score (Total), with values computed across five runs with different random seeds. We group the methods by category (generic SSL, single-cell contrastive SSL frameworks, generative methods, and baselines). For uni-modal data (PBMC, Pancreas, and Immune Cell Atlas), the specialized encoder-decoder method scVI, the domain-specific SSL method CLAIRE, and a foundation model scGPT outperform other methods. For the multi-modal datasets PBMC-M and BMMC, generic methods achieve higher scores.

| Method | PBMC-M | | | BMMC | | | PBMC | | | Pancreas | | | Immune Cell Atlas | | |
|---|---|---|---|---|---|---|---|---|---|---|---|---|---|---|---|
| | Bio | Batch | Total | Bio | Batch | Total | Bio | Batch | Total | Bio | Batch | Total | Bio | Batch | Total |
| SimCLR | 0.877 | 0.434 | 0.700 | **0.877** | 0.601 | **0.767** | 0.370 | 0.563 | 0.447 | 0.791 | 0.615 | 0.721 | 0.555 | 0.753 | 0.635 |
| | ± 0.020 | ± 0.001 | ± 0.012 | ± 0.025 | ± 0.002 | ± 0.016 | ± 0.002 | ± 0.005 | ± 0.003 | ± 0.002 | ± 0.019 | ± 0.009 | ± 0.020 | ± 0.016 | ± 0.017 |
| MoCo | 0.786 | **0.581** | 0.704 | 0.647 | **0.819** | 0.716 | 0.336 | 0.594 | 0.439 | 0.754 | 0.638 | 0.707 | 0.404 | **0.882** | 0.595 |
| | ± 0.005 | ± 0.016 | ± 0.003 | ± 0.048 | ± 0.024 | ± 0.038 | ± 0.007 | ± 0.014 | ± 0.010 | ± 0.008 | ± 0.017 | ± 0.011 | ± 0.024 | ± 0.022 | ± 0.016 |
| SimSiam | 0.903 | 0.455 | 0.724 | 0.753 | 0.571 | 0.680 | 0.271 | 0.512 | 0.368 | 0.531 | 0.635 | 0.572 | 0.358 | 0.640 | 0.470 |
| | ± 0.057 | ± 0.029 | ± 0.046 | ± 0.007 | ± 0.002 | ± 0.005 | ± 0.016 | ± 0.002 | ± 0.011 | ± 0.113 | ± 0.017 | ± 0.061 | ± 0.040 | ± 0.020 | ± 0.030 |
| NNCLR | 0.877 | 0.534 | 0.740 | 0.819 | 0.580 | 0.723 | 0.345 | 0.544 | 0.424 | 0.701 | 0.579 | 0.652 | 0.430 | 0.665 | 0.524 |
| | ± 0.033 | ± 0.004 | ± 0.018 | ± 0.021 | ± 0.008 | ± 0.016 | ± 0.011 | ± 0.009 | ± 0.010 | ± 0.052 | ± 0.015 | ± 0.037 | ± 0.028 | ± 0.007 | ± 0.017 |
| BYOL | **0.928** | 0.493 | **0.754** | 0.742 | 0.693 | 0.722 | 0.134 | 0.748 | 0.379 | 0.578 | 0.659 | 0.610 | 0.222 | 0.864 | 0.479 |
| | ± 0.065 | ± 0.016 | ± 0.033 | ± 0.043 | ± 0.016 | ± 0.019 | ± 0.017 | ± 0.076 | ± 0.020 | ± 0.029 | ± 0.012 | ± 0.013 | ± 0.031 | ± 0.009 | ± 0.021 |
| VICReg | 0.814 | 0.405 | 0.651 | 0.832 | 0.656 | 0.761 | 0.412 | 0.607 | 0.490 | 0.811 | 0.617 | **0.733** | 0.529 | 0.816 | 0.644 |
| | ± 0.039 | ± 0.026 | ± 0.013 | ± 0.051 | ± 0.009 | ± 0.027 | ± 0.010 | ± 0.000 | ± 0.006 | ± 0.003 | ± 0.001 | ± 0.002 | ± 0.014 | ± 0.022 | ± 0.012 |
| Barlow Twins | 0.902 | 0.430 | 0.713 | 0.859 | 0.612 | 0.760 | 0.341 | 0.523 | 0.414 | 0.694 | 0.580 | 0.648 | 0.535 | 0.734 | 0.614 |
| | ± 0.048 | ± 0.014 | ± 0.034 | ± 0.018 | ± 0.011 | ± 0.006 | ± 0.010 | ± 0.005 | ± 0.004 | ± 0.011 | ± 0.010 | ± 0.034 | ± 0.012 | ± 0.020 | |
| Concerto | 0.785 | 0.422 | 0.64 | 0.524 | 0.661 | 0.579 | 0.055 | 0.566 | 0.260 | 0.102 | 0.367 | 0.208 | 0.426 | 0.810 | 0.580 |
| | ± 0.002 | ± 0.003 | ± 0.002 | ± 0.019 | ± 0.008 | ± 0.015 | ± 0.000 | ± 0.000 | ± 0.000 | ± 0.003 | ± 0.000 | ± 0.002 | ± 0.014 | ± 0.025 | ± 0.017 |
| CLEAR | — | — | — | — | — | — | 0.580 | 0.209 | 0.432 | 0.698 | 0.249 | 0.518 | 0.775 | 0.327 | 0.596 |
| | | | | | | | ± 0.000 | ± 0.002 | ± 0.001 | ± 0.011 | ± 0.002 | ± 0.006 | ± 0.037 | ± 0.008 | ± 0.022 |
| CLAIRE | — | — | — | — | — | — | 0.714 | 0.866 | 0.774 | 0.582 | **0.959** | 0.732 | 0.548 | 0.527 | 0.539 |
| | | | | | | | ± 0.009 | ± 0.005 | ± 0.008 | ± 0.003 | ± 0.014 | ± 0.004 | ± 0.018 | ± 0.011 | ± 0.011 |
| scCLIP | 0.643 | 0.402 | 0.546 | 0.638 | 0.194 | 0.460 | | | | | | | | | |
| | ± 0.002 | ± 0.004 | ± 0.001 | ± 0.006 | ± 0.005 | ± 0.005 | | | | | | | | | |
| scGPT (zero-shot) | — | — | — | — | — | — | 0.440 | 0.469 | 0.451 | 0.473 | 0.168 | 0.351 | 0.380 | 0.516 | 0.435 |
| | | | | | | | ± 0.010 | ± 0.017 | ± 0.013 | ± 0.001 | ± 0.003 | ± 0.002 | ± 0.012 | ± 0.014 | ± 0.008 |
| scGPT (finetuned) | — | — | — | — | — | — | **0.940** | 0.514 | 0.770 | **0.873** | 0.345 | 0.662 | **0.979** | 0.485 | **0.781** |
| | | | | | | | ± 0.012 | ± 0.011 | ± 0.011 | ± 0.003 | ± 0.047 | ± 0.021 | ± 0.017 | ± 0.019 | ± 0.017 |
| Geneformer (finetuned) | — | — | — | — | — | — | 0.024 | 0.462 | 0.199 | 0.004 | 0.437 | 0.177 | 0.013 | 0.265 | 0.114 |
| | | | | | | | ± 0.000 | ± 0.000 | ± 0.000 | ± 0.000 | ± 0.000 | ± 0.000 | ± 0.000 | ± 0.000 | ± 0.000 |
| scButterfly | 0.702 | 0.391 | 0.577 | 0.781 | 0.297 | 0.587 | — | — | — | — | — | — | — | — | — |
| | ± 0.002 | ± 0.003 | ± 0.002 | ± 0.000 | ± 0.004 | ± 0.002 | | | | | | | | | |
| scTEL | 0.089 | 0.800 | 0.373 | 0.000 | 0.706 | 0.282 | — | — | — | — | — | — | — | — | — |
| | ± 0.002 | ± 0.004 | ± 0.001 | ± 0.006 | ± 0.005 | ± 0.005 | | | | | | | | | |
| totalVI / scVI | 0.702 | 0.305 | 0.543 | 0.755 | 0.272 | 0.562 | 0.918 | **0.871** | **0.899** | 0.805 | 0.511 | 0.688 | 0.862 | 0.593 | 0.754 |
| | ± 0.002 | ± 0.002 | ± 0.001 | ± 0.002 | ± 0.001 | ± 0.002 | ± 0.015 | ± 0.000 | ± 0.009 | ± 0.002 | ± 0.007 | ± 0.001 | ± 0.033 | ± 0.013 | ± 0.024 |
| SCDC | — | — | — | — | — | — | 0.679 | 0.605 | 0.649 | 0.648 | 0.350 | 0.529 | 0.698 | 0.565 | 0.645 |
| | | | | | | | ± 0.050 | ± 0.005 | ± 0.028 | ± 0.018 | ± 0.023 | ± 0.002 | ± 0.024 | ± 0.019 | ± 0.020 |
| PCA | 0.448 | 0.369 | 0.417 | 0.538 | 0.282 | 0.436 | 0.558 | 0.303 | 0.456 | 0.683 | 0.266 | 0.516 | 0.677 | 0.276 | 0.517 |
| | ± 0.000 | ± 0.001 | ± 0.000 | ± 0.002 | ± 0.007 | ± 0.002 | ± 0.003 | ± 0.000 | ± 0.002 | ± 0.001 | ± 0.000 | ± 0.001 | ± 0.000 | ± 0.000 | ± 0.000 |

batch correction while failing in bio conservation and scoring almost zero for both multi-modal datasets. Despite the success of scVI for uni-modal data, totalVI's results are unsatisfactory compared to the generic contrastive methods. For BMMC dataset, totalVI total score (0.562) is almost twice lower than then score of best method SimCLR (0.767).

**Cell Type Annotation:** We assess the cell-typing performance on the single-modal scRNA-seq datasets and CLEAR augmentations for the hold-out batch of the Immune Cell Atlas dataset (Figure 2) and all study data (Table H3). For all experiments, we do not use a projector during inference. Table H3 evaluates cell-typing for the Pancreas dataset

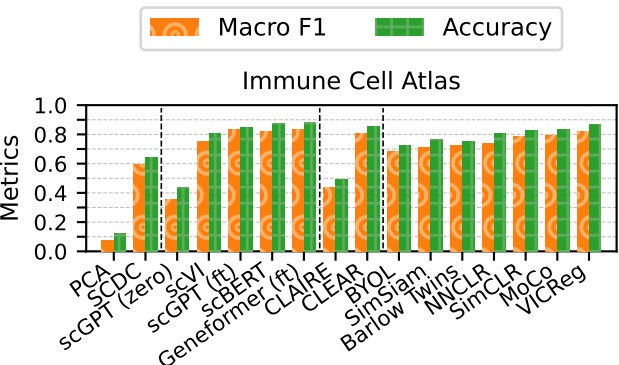

*Figure 2.* Uni-modal cell-typing with one sequencing technology (10X 5' v2) of the Immune Cell Atlas as a hold-out set. We train the encoder and classifier. The finetuned scGPT and Geneformer perform the best, while the generic VICReg method is a close third. The methods are grouped by category (baselines, specialized generative, specialized contrastive, and generic).

where unique batches were used as hold-out data. The best-performing methods are VICReg, CLEAR, and in rare cases finetuned single-cell foundation models (scFMs). All generic SSL methods perform well, together with scVI, which takes additional information as input. Although finetuned scFMs achieve adequate accuracy in most cases, they perform the best only for two datasets, the Immune Cell Atlas and the Xin study from the Pancreas, which are larger than the other three evaluated datasets. CLAIRE shows competitive results for the Pancreas dataset but falters in the second-biggest benchmarked dataset Immune Cell Atlas. Additionally, CLAIRE and scFMs have a significantly higher computational load than the other methods. PCA and zero-shot scGPT perform unsatisfactorily.

Table H4 shows cell-typing performance for multi-modal embeddings, where we integrate modalities through concatenation and train with CLEAR augmentations. We evaluate models using either both modalities (e.g., RNA and protein) or just the main modality (e.g., RNA) to assess whether representations capture information about the second modality and if a single main modality is sufficient during inference when the second modality is unavailable. scButterfly slightly outperforms VICReg and SimCLR, that show competitive results, with scButterfly leading overall performance (Table H4). All generic contrastive methods achieve good accuracy and outperform specialized contrastive methods like Concerto and scCLIP. While totalVI struggles with batch correction, it performs well in cell-typing. All models except scCLIP show better performance with multi-omics data than single modality, though the performance drop for single-modality inference is minimal. Notably, scCLIP appears to treat the second modality as noise. We conclude that scButterfly and generic contrastive models can be used for a single modality inference if the second modality is missing.

Of note, Concerto and totalVI do not support uni-modal inference for multi-modal data.

**Missing Modality Prediction:** Figure 3 shows the ability to predict missing protein values while given only RNA or gene expression (GEX) during inference. The model is trained on multi-omics data using CLEAR augmentations and concatenation to combine modalities. The standard deviation is close to zero, see Table H5. VICReg and SimCLR outperform other methods, including specialized single-cell frameworks. We assume that Concerto, scCLIP, and sc-TEL do not learn enough information about the secondary modality (protein) and its connection to the main modality (RNA) and, therefore, are not able to predict the missing modality. We evaluate scButterfly in two modes: averaging kNN, as done for other methods, and generating proteins directly from gene expression data. While the performance differences between the two approaches are insignificant, the scButterfly's Pearson correlation is unsatisfactory low compared to generic contrastive methods. High values of Pearson's correlations show that models effectively infer protein values from gene expression data.

**Summary and Findings:** Overall, specialized SSL methods designed for single-cell analysis have not demonstrated clear advantages over general-purpose approaches, except for scVI, finetuned scGPT, and CLAIRE. We attribute the superior performance of scVI, finetuned scGPT, and CLAIRE to the fact that these methods leverage experimental batch information during training. For multi-omics data, the best models are SimCLR and VICReg. Our findings indicate that current single-cell SSL methods such as scCLIP or totalVI need improvement for multi-modal downstream tasks, as they do not yet surpass or compete with generic architectures in performance.

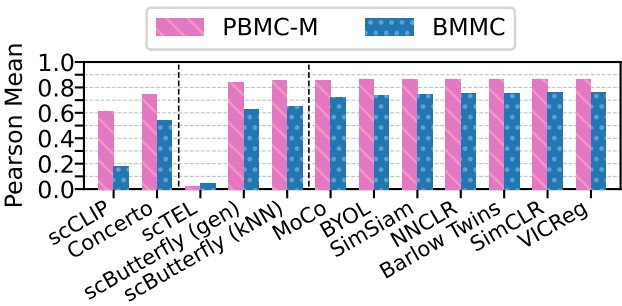

*Figure 3.* Missing modality prediction for models trained on the multi-modal datasets, PBMC and BMMC. We show the average Pearson correlation between the original and inferred missing modality: protein for PBMC-M and ADT (protein abundance) for BMMC. The methods are sorted from worst (left) to best (right) within group (specialized contrastive, generative, and generic).

## 4.2. Ablation Study

*RQ2* investigates how hyperparameters and augmentations impact the performance of single-cell SSL. We conduct hyperparameter tuning for all generic methods using two datasets: HIC and MCA. For the mentioned frameworks, we use the hyperparameters proposed in the respective original papers. For the generic methods, we focus on augmentations, the representation dimensionality, the projection dimensionality, and the temperature parameter.

**Representation Dimensionality:** We perform a grid search over the representation dimensionality for the HIC and MCA datasets, evaluating the batch correction performance (details in Appendix D). We train all models with embedding dimensions $\{8, 16, 32, 64, 128, 256, 512, 1024\}$. Models are ranked according to the SCIB (Luecken et al., 2022) total score, which is min-max scaled across all models. Our findings indicate that lower dimensionalities of 64 and 128 consistently lead to the best performance across all considered methods, while the larger dimensionality of 1024 achieves similar but requires more training time and memory (Figure G4 in Appendix). Given these observations, we adopt the embedding size of 64 for subsequent experiments.

**Projector Dimensionality:** To learn more robust representations, self-supervised models may benefit from projection heads (Xue et al., 2024). We investigate the impact of projection dimensionality during training by introducing a scale factor. At inference time, the projection head is discarded, and only the encoder is used. For contrastive methods, the projection size is scaled down by this factor, while for non-contrastive methods, it is scaled up by the same factor (see Appendix D). Our results reveal that the projector effect is ambiguous for most models (Figure G5). However, BarlowTwins, BYOL, and VICReg show an improved performance with larger scaling factors.

**Temperature Impact:** Figure 4 shows the temperature $t^\circ$ effect for SimCLR, MoCo, and NNCLR models. We evaluate $t^\circ \in \{0.1, 0.5, 1, 5, 10\}$ using SCIB-METRICS scores. Overall, lower $t^\circ$ values lead to better scores. The PyTorch default value ($t^\circ = 0.5$) performs well across all models and datasets (used for subsequent experiments). For scRNA-seq datasets, MCA and HIC, batch correction metric decreases with increasing $t^\circ$, except for NNCLR model and HIC dataset. For HIC data, higher $t^\circ$ leads to uncorrected batch effects, while for MCA data, it results in clustering of similar cell types that are expected to be near but not fully mixed together (e.g., CD4 T and CD8 T cells). For multi-modal datasets, PBMC-M and BMMC, $t^\circ$ effects are inconsistent. For PBMC-M, MoCo produces extremely similar results with all evaluated $t^\circ$, while SimCLR and NNCLR mix batches better with higher $t^\circ$, e.g., with a $t^\circ$ of 10 cell types are separated with mixed batches. For BMMC, lower $t^\circ$ achieves better batch correction.

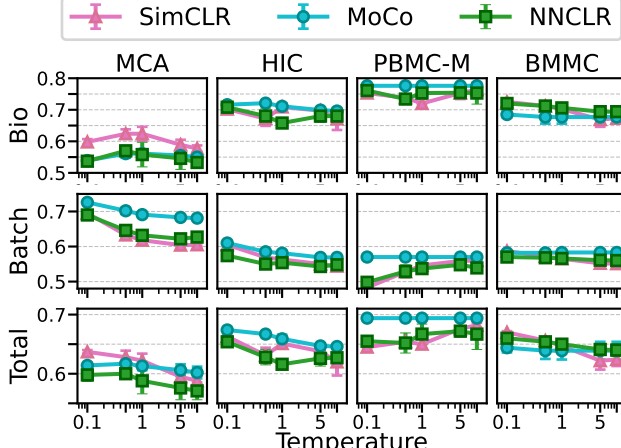

*Figure 4.* Temperature impact on the loss of three contrastive methods on four datasets (columns). Bio conservation, batch correction, and total scores are represented on the y-axis. The results are not min-max scaled for easier comparison. Overall, smaller temperature leads to better data integration.

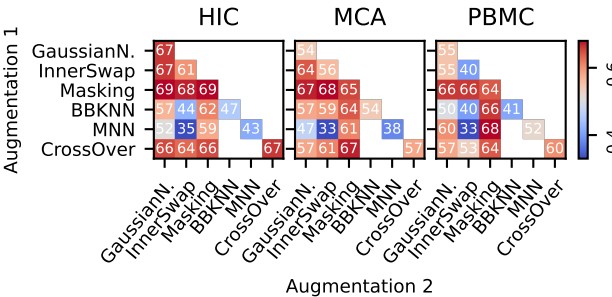

*Figure 5.* Evaluation of individual and combined data augmentations for the VICReg method based on total score for batch correction. Diagonal entries correspond to a single augmentation, and off-diagonal entries correspond to the two sequentially applied augmentations. Hyperparameters are based on ablation results (Table H10), evaluation for SimCLR and MoCo are in Figure G7.

**Augmentation Ablation:** The space of augmentations in the single-cell domain can be split into: random transformations (Han et al., 2022; Richter et al., 2024b) and neighborhood-based transformations (Yan et al., 2023; Liu et al., 2024). We perform an ablation for all studied augmentations and optimize hyperparameters for each (see Appendix E). To study how augmentations affect each other, we train VICReg, SimCLR, and MoCo models with combinations of two augmentations. We choose these models due to their consistently good performance. Random masking is the best-performing augmentation alone and combined with others (Figure 5). Additionally, CrossOver performs competitively, especially for the SimCLR model (Figure G7).

**Summary and Findings:** Higher-dimensional embeddings and lower temperatures enhance consistency and performance. Larger representations have better bio conservation,

smaller - batch correction. However, the optimal embedding size is 64 or 128. Among various data augmentation techniques, masking proves most effective and surpasses even sophisticated biology-specific approaches that incorporate batch information, such as MNN and BBKNN.

### 4.3. Impact of Batch Normalization and Multi-Modal Integration Proposed for Image Data

In the following, we address *RQ3* and study the impact of common SSL techniques, such as retaining the projector and domain-specific batch normalization (Chang et al., 2019).

**Retaining Projector:** To evaluate the impact of the projector during inference, we train a model consisting of an encoder and projector with CLEAR augmentations, and evaluate it with/without projection. Although it is common to only use the encoder during inference to create an embedding (Chen et al., 2020; Chen & He, 2020), the projector is also leveraged in the single-cell community and we compare the two approaches. First, we analyze how retaining the projection layer affects batch integration for single-modal data (Table H6). While using the projection layer slightly improves batch correction, it generally reduces biological conservation and total (which weights bio conservation more) scores. For example, the total score decreases from 0.625 to 0.608 for SimCLR on the MCA dataset. These findings suggest using only the encoder during inference, rather than the combined encoder and projector. The former better preserves biological signals despite slightly worse batch correction performance. Second, we observe a similar trend on multi-modal datasets as on scRNA-seq data (Table H7). The effects are less consistent and conclusive (scores change among models/datasets), and only MoCo's batch correction benefits from projection. Remarkably, Concerto uses encoder and projector during inference but has comparably unsatisfactory batch correction performance.

**Domain-Specific Batch Normalization (DSBN):** Inspired by Concerto (Yang et al., 2022b) and common practices from computer vision applications, we evaluate whether models benefit from DSBN (Chang et al., 2019). Although the Concerto (Yang et al., 2022b) manuscript discusses the usage of DSBN, the publicly available code does not apply DSBN. Therefore, we evaluate DSBN only for generic methods. Table H8 shows reduced total performance when leveraging DSBN compared to standard batch normalization. For HIC dataset, DSBN leads to slightly better batch correction but worse bio conservation. However, it is not the case for the MCA dataset.

**Multi-Modal Integration Methods:** In Table H9, we compare three ways to combine multiple modalities of a cell: element-wise addition of uni-modal embeddings (Yang et al., 2022b), concatenation of uni-modal embeddings, and multi-modal contrastive learning with the CLIP objective (Radford

et al., 2021; Xiong et al., 2023). For each modality, we train a model with CLEAR (Han et al., 2022) augmentationsand discard the projector during inference. See Appendix F for details. Table H9 shows that concatenation is the best integration method. Addition and concatenation show high results in bio conservation, while the CLIP-based approach performs better in batch correction.

**Summary and Findings:** Previously recommended techniques, such as keeping the projector or using DSBN, fail to enhance performance. For combining multiple modalities, concatenation turns out to be the most effective approach.

## 5. Conclusions

We introduced a comprehensive benchmark, scSSL-Bench, for self-supervised learning on uni- and multi-modal single-cell data. First, we observe that specialized single-cell SSL methods perform better than generic methods for uni-modal data and underperform for multi-modal data (*RQ1*). The best scRNA-seq single-modal data integration methods are scVI, CLAIRE, and the finetuned scGPT, all specialized for single-cell data. Generative scVI and the single-cell foundation model scGPT significantly outperform all SSL methods, while CLAIRE shows good scores only for a subset of the datasets. For multi-omics data, the generic methods Sim-CLR and VICReg perform the best and even outperform all other methods in the cell type annotation and missing modality prediction tasks for single-modal data. According to our findings, there is a need for improving existing and developing new multi-modal specialized SSL methods since current existing frameworks do not outperform generic architectures, and multi-modal integration turned out to be a more difficult task than uni-modal (*RQ1*). Second, we conclude that masking augmentation leads to the biggest improvements alone and in combination with other types of augmentations, and moderately-size to large embedding sizes lead to better results (*RQ2*). Third, retaining the projection head or applying domain-specific batch normalization do not significantly influence the scores and rather degrade the total data integration score by achieving a lower bio conservation and higher batch correction indicating primarily regularization potential (*RQ3*). Finally, our benchmark offers a standardized framework for assessing new SSL methods, enabling researchers to systematically evaluate and compare their approaches against established baselines.

## Acknowledgements

The authors thank Sebastian Baunsgaard, Dmitry Kobak, and Thomas Sutter for their insightful comments and constructive feedback on the manuscript. We thank the three anonymous reviewers for their thorough reviews, which significantly improved the quality of this paper.

Computational data analysis was performed at Leonhard Med secure trusted research environment at ETH Zurich and at the BIFOLD Hydra cluster.

FB is supported by the Swiss National Science Foundation (SNSF) (grant number 205321-207931).

## Impact Statement

This work provides a systematic benchmark of self-supervised learning (SSL) methods in single-cell genomics; we evaluate nineteen approaches across batch correction, cell typing, and missing modality prediction tasks. Our results offer practical guidelines for applying SSL to biological data, advancing computational tools for single-cell analysis. The implications span from biomedical research to personalized medicine, where improved data integration enables better diagnostic and therapeutic strategies. By standardizing evaluation protocols, this benchmark promotes reproducibility and cross-disciplinary collaboration.

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

# A. Generic Self-Supervised Methods

In contrastive SSL, different augmentations/modalities of the same instance are used to create positive pairs (i.e., similar examples), while pairs of distinct instances represent negative pairs (i.e., dissimilar examples) (Chen et al., 2020). Non-contrastive methods, also called negative-free contrastive learning, leverage only positive pairs (Cao et al., 2024a).

**Contrastive Methods:** A common framework for contrastive learning is SimCLR (Chen et al., 2020). Originally, SimCLR applies three image-data-specific augmentations to create positive/negative pairs and maximizes agreement between different augmented views via a temperature-scaled cross-entropy loss (NTXent) (Sohn, 2016) loss. MoCo (He et al., 2020) uses an additional momentum encoder for the augmented views. The key advantage of Moco is improved memory efficiency. NNCLR (Dwibedi et al., 2021) samples nearest neighbors of an instance to define positive pairs. Both MoCo and NNCLR contrast via InfoNCE (van den Oord et al., 2019) loss. SimCLR (Chen et al., 2020), MoCo (He et al., 2020), and NNCLR (Dwibedi et al., 2021) rely on positive and negative samples.

**Non-Contrastive Methods:** The emergence of non-contrastive methods was facilitated by an improved understanding of instabilities during model training. BYOL (Grill et al., 2020) uses online (trainable) and target (fixed) networks to train a representation. Online and target networks receive actual samples and augmented views respectively. The target network is updated using exponential moving averages of weights of the previous online networks. BYOL (Grill et al., 2020) minimizes the similarity of two representations produced by both networks. Additionally, using momentum encoders in MoCo (He et al., 2020) and BYOL (Grill et al., 2020) helps against dimensionality collapse from, for instance, the lack of negative pairs. Dimensionality collapse appears when embeddings span in a lower-dimensional subspace instead of the entire space (Jing et al., 2022). SimSiam, like SimCLR, uses an encoder with shared weights to process two augmented views. One encoder output (*left*) is fed to an additional predictor network before maximizing similarity, while the other output (*right*) is used directly. The encoder and predictor are updated with only *left* path gradients. SimSiam (Chen & He, 2020) and BYOL (Grill et al., 2020) use a predictor network to achieve better performance and avoid representation collapse without leveraging negative pairs. Barlow Twins (Zbontar et al., 2021) uses augmentations to train two identical networks and calculates a cross-correlation matrix (Zbontar et al., 2021) between the trained representations to reduce redundancy of the embeddings. VICReg (Bardes et al., 2022) is a joint embedding architecture with variance, invariance, and covariance regularization. The authors introduce regularization terms to the loss to control the variance of embeddings and decorrelate latent variables.

# B. Datasets

All datasets used in our benchmark are publicly available.

**Human Immune Cells (HIC):** This dataset comprises 33,506 cells and includes 12,303 genes from ten different donors assembled by Luecken et al.(2022) from five studies. One study derived cells from the human bone marrow and the other four from the human peripheral blood. There are 16 cell types annotated in the dataset. Availability: `https://doi.org/10.6084/m9.figshare.12420968.v8`

**Mouse Cell Atlas (MCA):** This dataset comprises 6,954 cells collected across two studies (Tran et al., 2020) with the first study consisting of 4,239 cells and the second batch containing 2,715 cells. Three different sequencing protocols were used. The harmonized dataset contains 51,817 genes and eleven cell types. Availability: `https://ndownloader.figshare.com/files/10351110` and `https://ndownloader.figshare.com/files/10760158`

**Peripheral Blood Mononuclear Cells (PBMC):** Collected by Ding et al.(2020), this dataset contains 30,449 cells from two patients and includes 33,694 genes. Cells were sequenced with seven different protocols (10x Chromium (v2), 10x Chromium (v3), Drop-seq, inDrops, Chromium (v3), Seq-Well, CEL-Seq2). We have made use of the annotations of nine unique cell types (D4+ T cell, Cytotoxic T cell, Natural killer cell, CD16+ monocyte, CD14+ monocyte, Megakaryocyte, B cell, Dendritic cell, Plasmacytoid dendritic cell) provided in the original study. Also, we removed the unassigned cells. Availability: `https://singlecell.broadinstitute.org/single_cell/study/SCP424/single-cell-comparison-pbmc-data`

**Pancreas:** This dataset was collected by Tran et al.(2020) combining five studies of the human pancreas. It comprises 14,767 cells, with 5,975 genes shared across all studies, sequenced by four scRNA-seq technologies (inDrop, CEL-Seq2, Smart-Seq2, SMARTer). The harmonized dataset contains 13 cell types (alpha, beta, ductal, acinar, delta, pancreatic stellate, pancreatic polypeptide, endothelial, macrophage, mast, epsilon, Schwann and T cell). Availability: `https://figshare.com/ndownloader/files/24539828`

**Lung:** This dataset contains 32,426 cells across 16 batches and two technologies (Drop-seq and 10x Chromium), assembled by Luecken et al.(2022) from three labs. The harmonized dataset includes 15,148 genes. The cells are derived from transplant patients and lung biopsies and are annotated as 17 cell types. Availability: `https://figshare.com/ndownloader/files/24539942`

**Immune Cell Atlas:** This dataset contains 329,762 cells and includes 36,398 genes across twelve batches and three different sequencing technologies (10x 5' v1, 10x 5' v2, 10x 3' v3), collected by Conde et al.(2022). The cells originate from 16 different tissues. The annotations include 35 fine-grain cell types. Availability: `https://datasets.cellxgene.cziscience.com/08f58b32-a01b-4300-8ebc-2b93c18f26f7.h5ad`

**Tabula Sapiens:** This dataset was collected by Jones et al.(2022) and contains 1,136,218 cells from 24 tissues and organs, sequenced with 10x 3' v3, 10x 5' v2, Smart-seq, and Smart-seq3 protocols. Tabula Sapiens is a molecular reference atlas for more than 400 cell types of the human body. Availability: `https://cellxgene.cziscience.com/collections/e5f58829-1a66-40b5-a624-9046778e74f5`

**Multi-modal Peripheral Blood Mononuclear Cells (PBMC-M):** This dataset was collected by Hao et al.(2021) with 161,764 cells across eight batches. For each cell, two modalities are available: RNA and protein. RNA has 18,702 genes, while the dimension of protein is 224. As a pre-processing step, we merge different T cell granularities, similar to the Concerto framework (Yang et al., 2022b). Availability: `https://atlas.fredhutch.org/data/nygc/multimodal/pbmc_multimodal.h5seurat`

**Multi-modal Bone Marrow Mononuclear Cells (BMMC).** This dataset was collected by Luecken et al.(2021) and contains 90,261 cells across thirteen batches and twelwe healthy human donors (Lance et al., 2022). Each cell has two modalities: Gene expression (GEX) and protein abundance (ADT). While GEX has 13,953 genes, the protein abundance dimension is 134. Pre-processing is the same as PBMC-M. Availability: `https://www.ncbi.nlm.nih.gov/geo/query/acc.cgi?acc=GSE194122`

## C. Evaluation Details

**Preprocessing:** All datasets are preprocessed using SCANPY (Wolf et al., 2018) normalize-total function, which scales the total counts per cell to 10,000, followed by log-transformation. We subsequently perform batch-aware feature selection to choose the 4,000 most highly-variable genes (HVGs) for further processing. For multi-modal PBMC-M and BMMC datasets, we select 2,000 HVGs contrary to 4,000 HVGs for the single modality datasets.

**Batch Correction:** The evaluated metrics are divided into two categories: those that measure the conservation of biological variance and that that measure the batch correction (Tran et al., 2020; Luecken et al., 2022). To evaluate conservation of biological variation, we calculate the isolated labels score, the Leiden NMI and ARI, the silhouette label score, and the cLISI metric. To evaluate batch correction, we calculate the graph connectivity, kBET per label, iLISI for each cell, the PCR comparison score, and the silhouette coefficient per batch. For details and definitions of the used evaluation metrics, as well as their implementation, we refer to (Luecken et al., 2022).

**Cell-Type Annotation and Missing Modality Prediction:** In the PBMC-M dataset, for cell-type annotation mapping and missing modality inference, we hold out batches *P3*, *P5*, and *P8*. In the BMMC dataset, for cell-type annotation mapping and missing modality inference, we hold out batches *s4d1*, *s4d8*, and *s4d9*. Similar to the approach of (Xu et al., 2021), we perform cell-type annotation by fitting a non-parametric supervised classifier (k-nearest neighbors (KNN) classifier with $k = 11$). For missing modality prediction, we fit a KNN classifier with $k = 5$, as in (Yang et al., 2022b).

## D. Hyperparameter Tuning

In all experiments, we use the augmentation pipeline proposed by CLEAR (Han et al., 2022) as a foundation, unless stated differently. Experiments described in this section were computed for all methods except Concerto. For the latter, we use the original model from (Yang et al., 2022b).

**Optimization:** All models in this benchmark, except Concerto, were trained with the Adam optimizer (Kingma & Ba, 2017). We use a stepwise learning rate schedule with base learning rate `1e-4` and fix the batch size at 256. When applicable, the memory bank size was set to 2048.

**Encoder Architecture:** We fix the encoder across all architectures and only perform a hyper-parameter search on the

dimensionality of the encoder output, i.e., the representation dimensionality. The encoder consists of a fully connected layer reducing the dimensionality to 128, followed by a ReLU activation and batch normalization. A further fully connected layer encodes the hidden representation to the representation dimension, followed by batch normalization.

**Projector Dimensionality:** Projection heads benefit self-supervised models in learning robust representations (Xue et al., 2024). At inference, the projection head is discarded, and only the (backbone) encoder is used for inference. All evaluated architectures subject to our evaluation include a projection head. We perform a hyperparameter search to find the best output dimension of the projector.

All projection heads were implemented as noted in the respective works. In their respective works, SimCLR, MoCo, SimSiam, and NNCLR are evaluated with projectors that retain or scale down the dimensionality of the representation. BarlowTwins, BYOL, and VICReg are evaluated with projectors that retain or scale up the dimensionality. We follow this rationale and search a grid of scaling factors $\{1, 2, 4\}$. To compute the projection dimensionality, the scaling factor is either divided (scale-down models) or multiplied (scale-up models) with the representation's dimension.

**Regularization Hyperparameters:** Variance-invariance-covariance regularization hyperparameters are used as is done in the original work. We evaluate a grid of parameters, where the invariance term and the variance term $\lambda, \alpha = \{5, 10, 25, 50\}$, while the invariance term $\beta$ is fixed to 1. We find that $\lambda$ and $\alpha$ fixed to 5 perform well across both ablation datasets.

**Augmentation Strength:** Augmentations are known to benefit SSL models in finding robust representations. Details of the evaluated augmentations are listed in Appendix E. We perform a grid search to optimize the hyperparameters for all augmentations. This includes $\alpha$ for all models, $\sigma$ for the Gaussian Noise augmentation, and the *KNN*-size for the nearest-neighbor-based transforms MNN and BBKNN. For each augmentation, the original CLEAR hyperparameters are fixed, and only the hyperparameters of the evaluated augmentation are adapted. For the ablation of BBKNN, we remove CrossOver, and replace it by BBKNN. Due to the implementation of MNN, we remove CrossOver and insert MNN at the front of the augmentation pipeline. Results of the ablation are recorded in Figure G6.

## E. Augmentations

We evaluate six augmentations in this work. For all, the parameter $\alpha$ defines the proportion of values affected by the transform. Augmentations are applied sequentially. Masking is performed by setting gene expressions to zero. Gaussian noise computes a noise vector computed from the normal distribution (with zero mean and standard deviation $\sigma$) and adds it to the input. InnerSwap switches expressions between genes within a cell, while CrossOver switches expressions of the same gene between two *random* cells.

The MNN augmentation refers to our implementation of CLAIRE's augmentation (Yan et al., 2023). For each cell, it computes an intra- and inter-batch neighborhood based on its mutual nearest neighbors. Then, views are computed by interpolating between neighbors. We do not filter cell-neighborhoods based on representation similarities during early stages of model training, as is done in the original work. This work introduces the BBKNN augmentation. It uses a non-trimmed batch-balanced KNN graph (Polański et al., 2019) to define a set of *#batches×KNN* neighbors for each cell. Views are computed by interpolating between neighbors. It differs from CLAIRE's concept in that it does not distinguish between intra- and inter-batch neighbors. While MNN always produces a view based on neighbors within and a view based on neighbors from outside the batch, this is not the case for BBKNN. Due to its implementation, the MNN augmentation is limited to be applied first in any augmentation pipeline. We refer to (Yan et al., 2023) for further detail on the interpolation process.

## F. Multi-Modal Setting

Recent developments in the single-cell analysis allow the measurement of multiple aspects of a cellular state. Data containing multiple modalities of a cell, e.g., RNA and protein, is called multi-omics. Existing self-supervised methods for single-cell data integration can be extended to the multimodal setting by combining views produced by specialized models for different modalities. We train two models for each modality; each model consists of an encoder and projector. As is common (He et al., 2020; Chen et al., 2020; Geiping et al., 2023), only the encoder is used to infer the integrated representation. However, in the single-cell community, the projector is also used during inference, and, therefore, we also evaluate whether projection during prediction improves performance. Additionally, there are various techniques to combine representations (Radford et al., 2021; Xiong et al., 2023; Yang et al., 2022b). We evaluate three approaches: Addition, concatenation, and CLIP.

**Encoder & Projector Embedding Evaluation:** Using CLEAR augmentations, we train two models for each modality, each

consisting of an encoder and projector. In Table H7, we compare data integration performance with and without a projector during inference. Interestingly, SimCLR benefits from projection, while VICReg performance degrades. We conclude that the effect of projection is inconsistent across models.

# G. Supplementary Figures

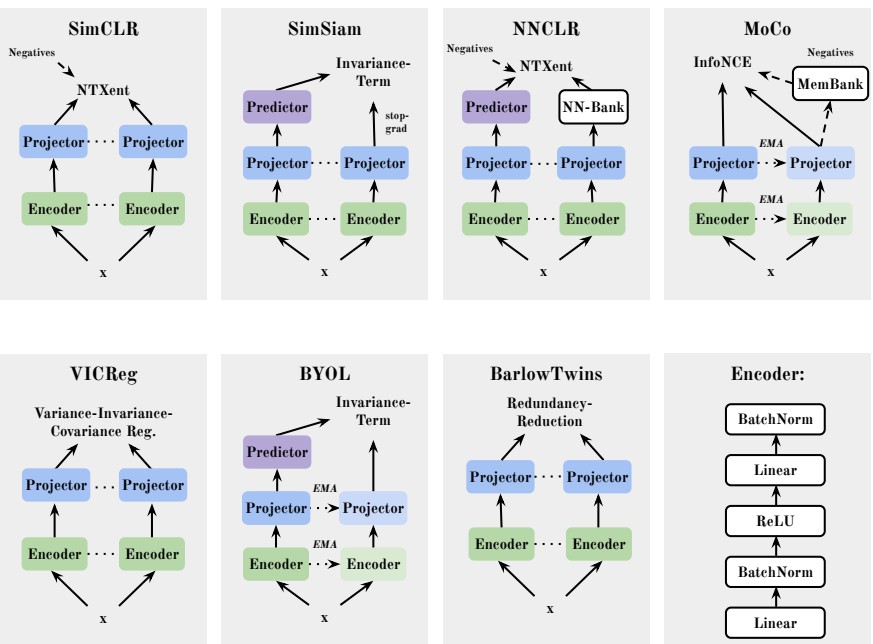

*Figure G1.* Overview of considered methods. Dotted lines between the encoder and projector blocks represent weight sharing. Exponential Moving Average (EMA) denotes the updating of weights with momentum. This figure was inspired by (Bardes et al., 2022; He et al., 2020; Dwibedi et al., 2021) based on our implementation of models with LightlySSL (Susmelj et al., 2023).

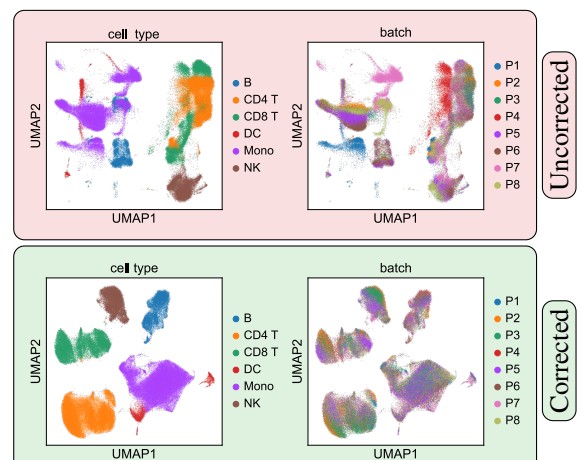

*Figure G2.* Batch Correction. The uncorrected (red) figure shows that cells cluster rather by batch (technical noise) than cell type (true biological signal) before the batch correction. After training a model and learning a corrected representation (green), cells are grouped by cell type, and batches are mixed.

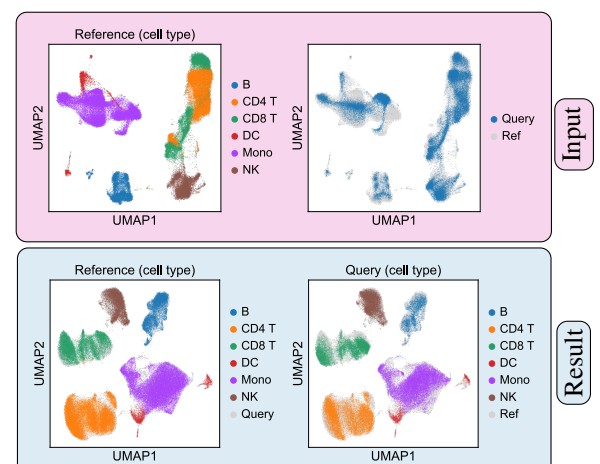

*Figure G3.* Query-to-Reference. Model gets an annotated train dataset (reference, pink input) as input and learns the corresponding latent space. During inference, representations of train (reference) and hold-out sets (query), train cell types, and a classifier are used to predict cell types of hold-out data (query, blue result).

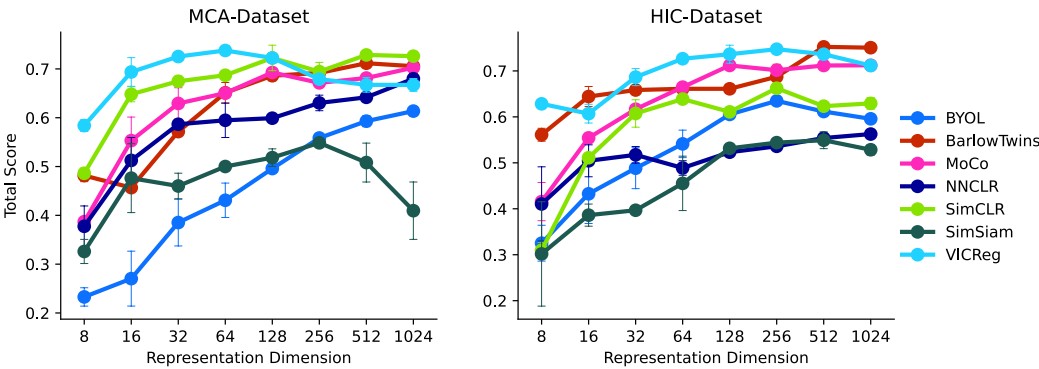

*Figure G4.* Tuning of the encoder based on the representation dimensionality. The encoder architecture is defined in Appendix D. Lines correspond to the mean total score across five runs with unique seeds.

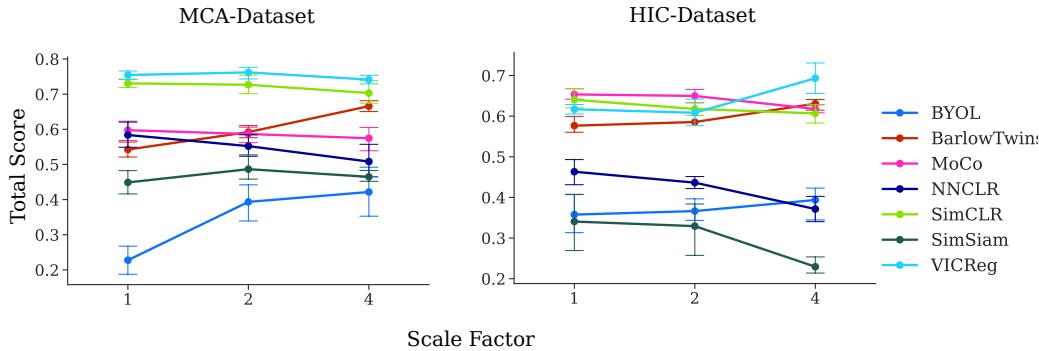

*Figure G5.* Tuning of the projector. The scale factor is defined in Appendix D: in contrastive methods, the projected size decreased according to the scale factor, while for non-contrastive methods the projection size increases in accordance with the scale factor. Lines correspond to the mean total score across five runs with unique seeds.

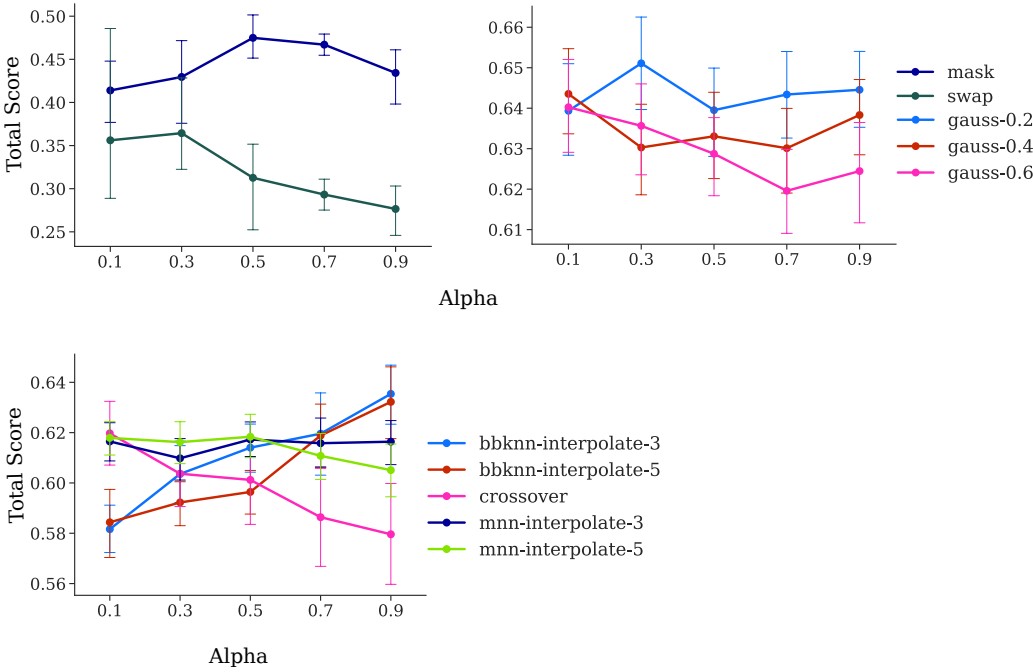

*Figure G6.* Ablation on the augmentation hyperparameters. The figure aggregates results for all methods, trained on the HIC dataset.

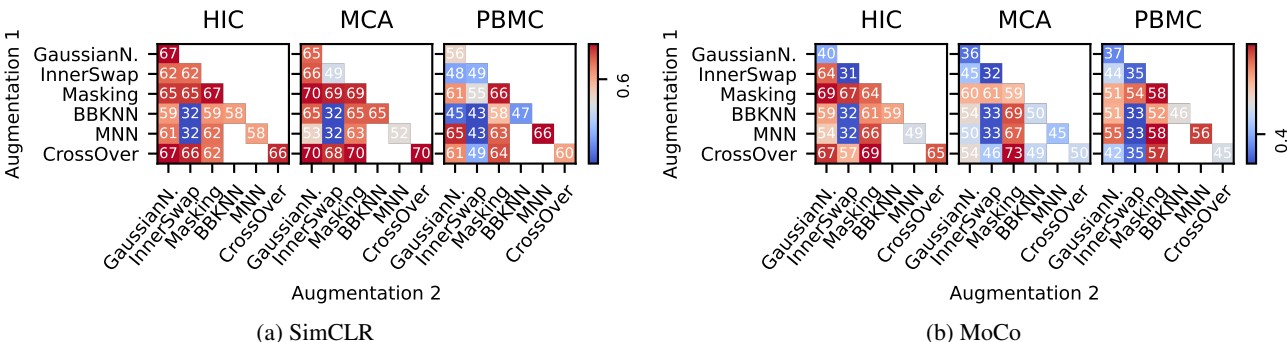

*Figure G7.* Evaluation of individual and combined data augmentations based on total score for batch correction for SimCLR and MoCo method. Diagonal entries correspond to a single augmentation, and off-diagonal entries correspond to the two sequentially applied augmentations.

## H. Supplementary Tables

*Table H1.* Batch correction benchmark for methods trained using the CLEAR pipeline. This table is an extension to Table 1, containing datasets that were used during hyperparameter tuning.

| Method | MCA | | | HIC | | | Lung | | |
|---|---|---|---|---|---|---|---|---|---|
| | Bio | Batch | Total | Bio | Batch | Total | Bio | Batch | Total |
| SimCLR | 0.519 | 0.666 | 0.578 | 0.753 | 0.536 | 0.666 | 0.184 | 0.628 | 0.362 |
| | ± 0.032 | ± 0.026 | ± 0.021 | ± 0.027 | ± 0.013 | ± 0.018 | ± 0.005 | ± 0.025 | ± 0.007 |
| MoCo | 0.280 | 0.791 | 0.484 | 0.791 | 0.603 | 0.716 | 0.148 | 0.678 | 0.360 |
| | ± 0.050 | ± 0.020 | ± 0.034 | ± 0.014 | ± 0.014 | ± 0.008 | ± 0.000 | ± 0.004 | ± 0.001 |
| SimSiam | 0.154 | 0.673 | 0.362 | 0.585 | 0.487 | 0.546 | 0.100 | 0.654 | 0.322 |
| | ± 0.043 | ± 0.033 | ± 0.036 | ± 0.043 | ± 0.055 | ± 0.038 | ± 0.019 | ± 0.055 | ± 0.011 |
| NNCLR | 0.332 | 0.665 | 0.465 | 0.695 | 0.500 | 0.617 | 0.147 | 0.662 | 0.353 |
| | ± 0.084 | ± 0.035 | ± 0.060 | ± 0.019 | ± 0.008 | ± 0.010 | ± 0.002 | ± 0.019 | ± 0.008 |
| BYOL | 0.000 | 0.694 | 0.277 | 0.582 | 0.707 | 0.632 | 0.028 | 0.705 | 0.299 |
| | ± 0.009 | ± 0.037 | ± 0.016 | ± 0.021 | ± 0.027 | ± 0.022 | ± 0.003 | ± 0.003 | ± 0.001 |
| VICReg | 0.515 | 0.709 | 0.592 | **0.830** | 0.581 | **0.730** | 0.208 | 0.622 | 0.374 |
| | ± 0.018 | ± 0.014 | ± 0.011 | ± 0.024 | ± 0.013 | ± 0.016 | ± 0.008 | ± 0.003 | ± 0.004 |
| Barlow Twins | 0.471 | 0.627 | 0.533 | 0.784 | 0.533 | 0.684 | 0.173 | 0.651 | 0.364 |
| | ± 0.032 | ± 0.029 | ± 0.026 | ± 0.011 | ± 0.004 | ± 0.008 | ± 0.007 | ± 0.010 | ± 0.000 |
| Concerto | 0.497 | 0.559 | 0.522 | 0.000 | 0.627 | 0.251 | 0.152 | 0.711 | 0.376 |
| | ± 0.036 | ± 0.013 | ± 0.027 | ± 0.000 | ± 0.004 | ± 0.001 | ± 0.008 | ± 0.008 | ± 0.008 |
| CLEAR | **0.684** | 0.385 | 0.565 | 0.615 | 0.262 | 0.474 | **0.876** | 0.326 | 0.656 |
| | ± 0.042 | ± 0.008 | ± 0.027 | ± 0.004 | ± 0.007 | ± 0.005 | ± 0.088 | ± 0.049 | ± 0.050 |
| CLAIRE | 0.467 | **0.978** | **0.672** | 0.511 | **0.974** | 0.696 | 0.315 | **0.907** | 0.552 |
| | ± 0.015 | ± 0.007 | ± 0.010 | ± 0.024 | ± 0.000 | ± 0.014 | ± 0.324 | ± 0.058 | ± 0.176 |
| scVI | 0.723 | 0.254 | 0.536 | 0.768 | 0.659 | 0.725 | 0.716 | 0.659 | **0.693** |
| | ± 0.021 | ± 0.017 | ± 0.019 | ± 0.007 | ± 0.003 | ± 0.006 | ± 0.176 | ± 0.017 | ± 0.111 |
| PCA | 0.538 | 0.297 | 0.442 | 0.635 | 0.164 | 0.446 | 0.748 | 0.245 | 0.547 |
| | ± 0.027 | ± 0.000 | ± 0.016 | ± 0.014 | ± 0.003 | ± 0.009 | ± 0.114 | ± 0.062 | ± 0.093 |

*Table H2.* Uni-modal cell-typing with CLEAR augmentations. We define one technology (10X 5' v2) of the Immune Cell Atlas as a hold-out set, train the encoder and knn-classifier. The generic model VICReg outperforms all other methods.

| Method | Immune Cell Atlas (10X 5' v2) | |
| --- | --- | --- |
| | Macro F1 | Acc |
| SimCLR | 0.788 ± 0.004 | 0.830 ± 0.003 |
| MoCo | 0.794 ± 0.006 | 0.835 ± 0.014 |
| SimSiam | 0.711 ± 0.015 | 0.768 ± 0.007 |
| NNCLR | 0.740 ± 0.010 | 0.804 ± 0.007 |
| BYOL | 0.680 ± 0.002 | 0.724 ± 0.009 |
| VICReg | 0.820 ± 0.012 | 0.866 ± 0.003 |
| Barlow Twins | 0.727 ± 0.004 | 0.752 ± 0.007 |
| CLEAR | 0.806 ± 0.000 | 0.855 ± 0.000 |
| CLAIRE | 0.436 ± 0.000 | 0.492 ± 0.002 |
| scGPT (zero-shot) | 0.358 ± 0.004 | 0.439 ± 0.001 |
| scGPT (finetuned) | **0.835** ± 0.008 | 0.850 ± 0.008 |
| Geneformer (finetuned) | 0.831 ± 0.000 | **0.878** ± 0.000 |
| scBERT | 0.818 ± 0.009 | 0.873 ± 0.004 |
| scVI | 0.750 ± 0.001 | 0.804 ± 0.000 |
| SCDC | 0.595 ± 0.009 | 0.642 ± 0.000 |
| PCA | 0.071 ± 0.007 | 0.124 ± 0.000 |

*Table H3.* Cell-type annotation with CLEAR augmentations on the Pancreas dataset. We define individual studies as holdout sets during training. Accuracy and Macro F1 are computed on the holdout set.

| Method | Mutaro et al. | | Segerstolpe et al. | | Wang et al. | | Xin et al. | |
|---|---|---|---|---|---|---|---|---|
| | Macro F1 | Acc | Macro F1 | Acc | Macro F1 | Acc | Macro F1 | Acc |
| SimCLR | 0.796 | 0.936 | 0.781 | 0.946 | **0.894** | **0.945** | 0.778 | 0.819 |
| | ± 0.014 | ± 0.012 | ± 0.022 | ± 0.011 | ± 0.005 | ± 0.010 | ± 0.050 | ± 0.019 |
| MoCo | 0.825 | 0.939 | 0.799 | 0.936 | 0.844 | 0.919 | 0.722 | 0.798 |
| | ± 0.045 | ± 0.013 | ± 0.033 | ± 0.006 | ± 0.020 | ± 0.021 | ± 0.045 | ± 0.010 |
| SimSiam | 0.679 | 0.875 | 0.595 | 0.803 | 0.586 | 0.798 | 0.464 | 0.718 |
| | ± 0.042 | ± 0.026 | ± 0.022 | ± 0.014 | ± 0.036 | ± 0.022 | ± 0.071 | ± 0.037 |
| NNCLR | 0.707 | 0.897 | 0.619 | 0.854 | 0.729 | 0.868 | 0.472 | 0.721 |
| | ± 0.043 | ± 0.024 | ± 0.013 | ± 0.012 | ± 0.019 | ± 0.012 | ± 0.033 | ± 0.025 |
| BYOL | 0.720 | 0.884 | 0.670 | 0.854 | 0.656 | 0.813 | 0.524 | 0.724 |
| | ± 0.037 | ± 0.021 | ± 0.005 | ± 0.006 | ± 0.048 | ± 0.008 | ± 0.062 | ± 0.013 |
| VICReg | 0.853 | 0.947 | 0.855 | **0.976** | 0.877 | 0.937 | 0.830 | 0.839 |
| | ± 0.039 | ± 0.002 | ± 0.008 | ± 0.006 | ± 0.008 | ± 0.004 | ± 0.023 | ± 0.010 |
| Barlow Twins | 0.700 | 0.868 | 0.673 | 0.870 | 0.737 | 0.878 | 0.487 | 0.720 |
| | ± 0.010 | ± 0.005 | ± 0.017 | ± 0.007 | ± 0.012 | ± 0.010 | ± 0.003 | ± 0.009 |
| Concerto | 0.106 | 0.431 | 0.113 | 0.419 | 0.105 | 0.435 | 0.112 | 0.406 |
| | ± 0.000 | ± 0.000 | ± 0.000 | ± 0.000 | ± 0.000 | ± 0.000 | ± 0.000 | ± 0.000 |
| CLEAR | **0.950** | **0.961** | 0.898 | 0.967 | 0.891 | 0.941 | 0.987 | 0.994 |
| | ± 0.000 | ± 0.001 | ± 0.000 | ± 0.002 | ± 0.000 | ± 0.007 | ± 0.000 | ± 0.001 |
| CLAIRE | 0.941 | 0.937 | **0.919** | 0.955 | 0.893 | 0.945 | 0.965 | 0.989 |
| | ± 0.000 | ± 0.002 | ± 0.000 | ± 0.002 | ± 0.000 | ± 0.001 | ± 0.000 | ± 0.002 |
| scGPT (zero-shot) | 0.502 | 0.765 | 0.549 | 0.826 | 0.327 | 0.519 | 0.581 | 0.786 |
| | ± 0.00 | ± 0.00 | ± 0.02 | ± 0.01 | ± 0.15 | ± 0.21 | ± 0.00 | ± 0.00 |
| scGPT (finetuned) | 0.850 | 0.917 | 0.803 | 0.955 | 0.466 | 0.560 | **0.989** | **0.995** |
| | ± 0.001 | ± 0.003 | ± 0.02 | ± 0.010 | ± 0.154 | ± 0.217 | ± 0.006 | ± 0.002 |
| Geneformer (finetuned) | 0.622 | 0.916 | 0.630 | 0.944 | 0.673 | 0.891 | 0.564 | 0.993 |
| | ± 0.000 | ± 0.000 | ± 0.000 | ± 0.000 | ± 0.000 | ± 0.000 | ± 0.000 | ± 0.000 |
| scBERT | 0.642 | 0.919 | 0.715 | 0.953 | 0.710 | 0.902 | 0.975 | 0.990 |
| | ± 0.004 | ± 0.012 | ± 0.002 | ± 0.002 | ± 0.0830 | ± 0.001 | ± 0.007 | ± 0.000 |
| scVI | 0.616 | 0.889 | 0.635 | 0.903 | 0.680 | 0.897 | 0.741 | 0.916 |
| | ± 0.015 | ± 0.012 | ± 0.127 | ± 0.023 | ± 0.135 | ± 0.023 | ± 0.171 | ± 0.023 |
| SCDC | 0.571 | 0.860 | 0.581 | 0.875 | 0.689 | 0.908 | 0.442 | 0.800 |
| | ± 0.004 | ± 0.021 | ± 0.001 | ± 0.015 | ± 0.003 | ± 0.002 | ± 0.002 | ± 0.009 |
| PCA | 0.071 | 0.082 | 0.171 | 0.236 | 0.195 | 0.371 | 0.054 | 0.427 |
| | ± 0.000 | ± 0.00 | ± 0.000 | ± 0.000 | ± 0.000 | ± 0.000 | ± 0.000 | ± 0.000 |

*Table H4.* Cell-type annotation for multi-modal datasets with CLEAR pipeline. On the left, two modalities (RNA + Protein or GEX (gene expression) + ADT (protein abundance)) were used during inference. On the right, we show inference performance with a single modality (RNA or GEX). All models were trained with two modalities.

| Method | RNA + Protein | | GEX + ADT | | RNA | | GEX | |
| --- | --- | --- | --- | --- | --- | --- | --- | --- |
| | PBMC-M | | BMMC | | PBMC-M | | BMMC | |
| | Macro F1 | Acc | Macro F1 | Acc | Macro F1 | Acc | Macro F1 | Acc |
| SimCLR | **0.950** | **0.977** | 0.770 | 0.876 | 0.906 | 0.946 | 0.749 | 0.848 |
| | ± 0.002 | ± 0.001 | ± 0.028 | ± 0.022 | ± 0.002 | ± 0.001 | ± 0.050 | ± 0.035 |
| MoCo | 0.930 | 0.969 | 0.609 | 0.771 | 0.778 | 0.835 | 0.630 | 0.717 |
| | ± 0.007 | ± 0.004 | ± 0.001 | ± 0.041 | ± 0.016 | ± 0.010 | ± 0.065 | ± 0.073 |
| SimSiam | 0.933 | 0.968 | 0.666 | 0.820 | 0.846 | 0.884 | 0.670 | 0.792 |
| | ± 0.002 | ± 0.001 | ± 0.069 | ± 0.036 | ± 0.016 | ± 0.019 | ± 0.082 | ± 0.056 |
| NNCLR | 0.941 | 0.971 | 0.734 | 0.856 | 0.860 | 0.901 | 0.703 | 0.806 |
| | ± 0.004 | ± 0.002 | ± 0.055 | ± 0.043 | ± 0.008 | ± 0.003 | ± 0.085 | ± 0.070 |
| BYOL | 0.933 | 0.968 | 0.737 | 0.847 | 0.857 | 0.898 | 0.704 | 0.795 |
| | ± 0.008 | ± 0.004 | ± 0.051 | ± 0.037 | ± 0.047 | ± 0.034 | ± 0.048 | ± 0.054 |
| VICReg | **0.950** | **0.977** | 0.808 | 0.899 | 0.923 | 0.957 | 0.785 | 0.887 |
| | ± 0.001 | ± 0.000 | ± 0.021 | ± 0.017 | ± 0.005 | ± 0.004 | ± 0.014 | ± 0.015 |
| Barlow Twins | 0.949 | 0.976 | 0.766 | 0.863 | 0.890 | 0.919 | 0.733 | 0.816 |
| | ± 0.001 | ± 0.001 | ± 0.012 | ± 0.003 | ± 0.014 | ± 0.008 | ± 0.013 | ± 0.017 |
| Concerto | 0.892 | 0.947 | 0.673 | 0.825 | — | — | — | — |
| | ± 0.000 | ± 0.001 | ± 0.000 | ± 0.000 | | | | |
| scCLIP | 0.699 | 0.851 | 0.557 | 0.797 | 0.728 | 0.857 | 0.635 | 0.818 |
| | ± 0.009 | ± 0.025 | ± 0.015 | ± 0.008 | ± 0.019 | ± 0.004 | ± 0.071 | ± 0.051 |
| scButterfly | 0.949 | 0.976 | **0.844** | **0.924** | **0.946** | **0.973** | **0.831** | **0.920** |
| | ± 0.000 | ± 0.000 | ± 0.002 | ± 0.000 | ± 0.000 | ± 0.000 | ± 0.000 | ± 0.001 |
| scTEL | 0.173 | 0.211 | 0.039 | 0.149 | — | — | —- | — |
| | ± 0.001 | ± 0.031 | ± 0.003 | ± 0.004 | | | | |
| totalVI | 0.829 | **0.977** | 0.829 | 0.911 | — | — | — | — |
| | ± 0.021 | ± 0.0158 | ± 0.023 | ± 0.015 | | | | |

*Table H5.* Missing modality prediction for methods trained with the CLEAR pipeline on multi-modal datasets. We show the average Pearson correlation between the original and inferred missing modality: protein for PBMC-M, and ADT (protein abundance) for BMMC.

| Method | PBMC-M Pearson Mean | BMMC Pearson Mean |
|---|---|---|
| SimCLR | **0.866** ± 0.001 | 0.757 ± 0.002 |
| MoCo | 0.856 ± 0.001 | 0.721 ± 0.004 |
| SimSiam | 0.859 ± 0.002 | 0.748 ± 0.002 |
| NNCLR | 0.861 ± 0.002 | 0.751 ± 0.001 |
| BYOL | 0.860 ± 0.000 | 0.738 ± 0.002 |
| VICReg | 0.865 ± 0.001 | **0.759** ± 0.001 |
| Barlow Twins | 0.864 ± 0.001 | 0.755 ± 0.001 |
| Concerto | 0.742 ± 0.006 | 0.542 ± 0.001 |
| scCLIP | 0.614 ± 0.003 | 0.175 ± 0.005 |
| scButterfly (kNN) | 0.856 ± 0.000 | 0.651 ± 0.001 |
| scButterfly (generated) | 0.840 ± 0.000 | 0.624 ± 0.002 |
| scTEL (100 epochs) | 0.022 ± 0.005 | 0.047 ± 0.002 |

*Table H6.* Batch correction results for methods with and without projection layer during inference. All methods are trained using the CLEAR pipeline. Results are not min-max scaled for easier comparison.

| Method | Encoder | | | | | | Encoder + Projection | | | | | |
|---|---|---|---|---|---|---|---|---|---|---|---|---|
| | | MCA | | | HIC | | | MCA | | | HIC | |
| | Bio | Batch | Total | Bio | Batch | Total | Bio | Batch | Total | Bio | Batch | Total |
| SimCLR | 0.620 ± 0.020 | 0.633 ± 0.007 | 0.625 ± 0.014 | 0.683 ± 0.008 | 0.567 ± 0.001 | 0.637 ± 0.004 | 0.575 ± 0.003 | 0.658 ± 0.006 | 0.608 ± 0.000 | 0.674 ± 0.010 | 0.575 ± 0.009 | 0.635 ± 0.002 |
| MoCo | 0.561 ± 0.013 | 0.706 ± 0.002 | 0.619 ± 0.007 | **0.721** ± 0.000 | 0.585 ± 0.003 | 0.667 ± 0.001 | 0.536 ± 0.000 | **0.726** ± 0.015 | 0.612 ± 0.006 | 0.705 ± 0.019 | 0.602 ± 0.001 | 0.664 ± 0.012 |
| SimSiam | 0.519 ± 0.008 | 0.656 ± 0.025 | 0.574 ± 0.006 | 0.623 ± 0.040 | 0.545 ± 0.012 | 0.592 ± 0.029 | 0.435 ± 0.009 | 0.681 ± 0.027 | 0.533 ± 0.006 | 0.598 ± 0.038 | 0.540 ± 0.017 | 0.575 ± 0.030 |
| NNCLR | 0.570 ± 0.012 | 0.646 ± 0.003 | 0.600 ± 0.008 | 0.680 ± 0.023 | 0.550 ± 0.003 | 0.628 ± 0.013 | 0.493 ± 0.001 | 0.675 ± 0.013 | 0.566 ± 0.004 | 0.657 ± 0.026 | 0.557 ± 0.005 | 0.617 ± 0.018 |
| BYOL | 0.486 ± 0.017 | 0.670 ± 0.003 | 0.560 ± 0.012 | 0.637 ± 0.007 | 0.602 ± 0.000 | 0.623 ± 0.004 | 0.398 ± 0.004 | 0.667 ± 0.028 | 0.506 ± 0.013 | 0.552 ± 0.010 | **0.607** ± 0.007 | 0.574 ± 0.009 |
| VICReg | 0.624 ± 0.005 | 0.653 ± 0.009 | 0.636 ± 0.006 | 0.716 ± 0.007 | 0.584 ± 0.002 | 0.663 ± 0.003 | 0.587 ± 0.002 | 0.699 ± 0.009 | **0.632** ± 0.005 | **0.726** ± 0.005 | 0.588 ± 0.005 | **0.671** ± 0.005 |
| Barlow Twins | 0.606 ± 0.009 | 0.623 ± 0.013 | 0.613 ± 0.000 | 0.716 ± 0.006 | 0.566 ± 0.006 | 0.656 ± 0.001 | 0.544 ± 0.018 | 0.664 ± 0.004 | 0.592 ± 0.012 | 0.675 ± 0.029 | 0.562 ± 0.006 | 0.630 ± 0.020 |
| Concerto | — | — | — | — | — | — | 0.634 ± 0.002 | 0.529 ± 0.000 | 0.592 ± 0.001 | 0.357 ± 0.000 | 0.470 ± 0.001 | 0.402 ± 0.001 |
| CLEAR | **0.696** ± 0.010 | 0.438 ± 0.001 | 0.593 ± 0.006 | 0.642 ± 0.003 | 0.408 ± 0.006 | 0.549 ± 0.001 | — | — | — | — | — | — |
| CLAIRE | 0.689 ± 0.018 | **0.763** ± 0.008 | **0.718** ± 0.014 | 0.699 ± 0.029 | **0.700** ± 0.003 | **0.699** ± 0.018 | — | — | — | — | — | — |
| PCA | 0.651 ± 0.001 | 0.348 ± 0.000 | 0.530 ± 0.000 | 0.654 ± 0.008 | 0.320 ± 0.000 | 0.521 ± 0.005 | **0.651** ± 0.001 | 0.348 ± 0.000 | 0.530 ± 0.000 | 0.654 ± 0.008 | 0.302 ± 0.000 | 0.521 ± 0.005 |

*Table H7.* Data integration for methods using the CLEAR pipeline on multimodal datasets. We compare the effect of retaining the projection head during inference to the representation quality when using only the encoder. This table is not min-max scaled.

| | Encoder | | | | | | Encoder + Projection | | | | | |
| | PBMC-M | | | BMMC | | | PBMC-M | | | BMMC | | |
| **Method** | Bio | Batch | Total | Bio | Batch | Total | Bio | Batch | Total | Bio | Batch | Total |
|---|---|---|---|---|---|---|---|---|---|---|---|---|
| SimCLR | 0.738 | 0.518 | 0.650 | 0.718 | 0.574 | 0.660 | 0.741 | 0.513 | 0.650 | 0.712 | 0.567 | 0.654 |
| | ± 0.000 | ± 0.010 | ± 0.004 | ± 0.016 | ± 0.001 | ± 0.010 | ± 0.021 | ± 0.005 | ± 0.015 | ± 0.004 | ± 0.001 | ± 0.002 |
| MoCo | 0.776 | **0.570** | 0.694 | 0.677 | **0.582** | 0.639 | 0.762 | **0.606** | **0.700** | 0.592 | **0.637** | 0.610 |
| | ± 0.001 | ± 0.007 | ± 0.002 | ± 0.021 | ± 0.004 | ± 0.014 | ± 0.009 | ± 0.004 | ± 0.004 | ± 0.025 | ± 0.007 | ± 0.012 |
| SimSiam | 0.766 | 0.563 | 0.685 | 0.677 | 0.559 | 0.629 | 0.767 | 0.565 | 0.686 | 0.635 | 0.553 | 0.602 |
| | ± 0.051 | ± 0.012 | ± 0.036 | ± 0.011 | ± 0.000 | ± 0.006 | ± 0.024 | ± 0.006 | ± 0.017 | ± 0.021 | ± 0.002 | ± 0.014 |
| NNCLR | 0.750 | 0.535 | 0.664 | 0.706 | 0.565 | 0.650 | 0.743 | 0.533 | 0.659 | 0.695 | 0.575 | 0.647 |
| | ± 0.007 | ± 0.010 | ± 0.000 | ± 0.002 | ± 0.001 | ± 0.002 | ± 0.032 | ± 0.024 | ± 0.029 | ± 0.014 | ± 0.006 | ± 0.011 |
| BYOL | **0.789** | 0.557 | **0.696** | 0.701 | 0.574 | 0.651 | **0.783** | 0.543 | 0.687 | 0.691 | 0.584 | 0.648 |
| | ± 0.015 | ± 0.000 | ± 0.009 | ± 0.005 | ± 0.007 | ± 0.006 | ± 0.007 | ± 0.009 | ± 0.008 | ± 0.006 | ± 0.002 | ± 0.004 |
| VICReg | 0.749 | 0.478 | 0.641 | **0.722** | **0.582** | 0.666 | 0.763 | 0.491 | 0.654 | **0.714** | 0.578 | **0.660** |
| | ± 0.008 | ± 0.017 | ± 0.002 | ± 0.002 | ± 0.001 | ± 0.000 | ± 0.001 | ± 0.003 | ± 0.002 | ± 0.000 | ± 0.001 | ± 0.001 |
| Barlow Twins | 0.755 | 0.509 | 0.657 | 0.704 | 0.577 | 0.653 | 0.712 | 0.506 | 0.630 | 0.704 | 0.581 | 0.655 |
| | ± 0.018 | ± 0.015 | ± 0.017 | ± 0.001 | ± 0.005 | ± 0.003 | ± 0.006 | ± 0.003 | ± 0.005 | ± 0.002 | ± 0.009 | ± 0.002 |
| Concerto | — | — | — | — | — | — | 0.773 | 0.436 | 0.638 | 0.604 | 0.525 | 0.573 |
| | — | — | — | — | — | — | ± 0.117 | ± 0.006 | ± 0.072 | ± 0.089 | ± 0.01 | ± 0.054 |
| PCA | 0.602 | 0.504 | 0.563 | 0.558 | 0.322 | 0.464 | 0.602 | 0.504 | 0.563 | 0.558 | 0.322 | 0.464 |
| | ± 0.000 | ± 0.000 | ± 0.000 | ± 0.000 | ± 0.000 | ± 0.000 | ± 0.000 | ± 0.000 | ± 0.000 | ± 0.000 | ± 0.000 | ± 0.000 |

*Table H8.* Batch correction benchmark for methods trained using the CLEAR pipeline with domain specific batch normalization (DSBN). Results are not min-max scaled for easier comparison.

| Method | Batch Normalization | | | | | | DSBN | | | | | |
| | HIC | | | MCA | | | HIC | | | MCA | | |
| | Bio | Batch | Total | Bio | Batch | Total | Bio | Batch | Total | Bio | Batch | Total |
|---|---|---|---|---|---|---|---|---|---|---|---|---|
| SimCLR | 0.703 | 0.573 | 0.651 | 0.627 | 0.644 | 0.634 | 0.680 | 0.583 | 0.641 | 0.624 | 0.636 | 0.629 |
| | ± 0.022 | ± 0.014 | ± 0.009 | ± 0.009 | ± 0.026 | ± 0.014 | ± 0.020 | ± 0.015 | ± 0.008 | ± 0.008 | ± 0.009 | ± 0.006 |
| MoCo | 0.707 | 0.582 | 0.657 | 0.518 | **0.731** | 0.603 | 0.648 | **0.612** | 0.633 | 0.549 | **0.697** | 0.608 |
| | ± 0.010 | ± 0.020 | ± 0.011 | ± 0.060 | ± 0.032 | ± 0.047 | ± 0.040 | ± 0.004 | ± 0.022 | ± 0.013 | ± 0.003 | ± 0.008 |
| SimSiam | 0.619 | 0.544 | 0.589 | 0.523 | 0.668 | 0.581 | 0.603 | 0.595 | 0.600 | 0.502 | 0.635 | 0.555 |
| | ± 0.053 | ± 0.043 | ± 0.049 | ± 0.069 | ± 0.072 | ± 0.069 | ± 0.074 | ± 0.025 | ± 0.034 | ± 0.018 | ± 0.013 | ± 0.015 |
| NNCLR | 0.658 | 0.546 | 0.613 | 0.574 | 0.637 | 0.599 | 0.659 | 0.590 | 0.632 | 0.543 | 0.651 | 0.587 |
| | ± 0.015 | ± 0.011 | ± 0.010 | ± 0.120 | ± 0.056 | ± 0.087 | ± 0.011 | ± 0.006 | ± 0.009 | ± 0.019 | ± 0.001 | ± 0.011 |
| BYOL | 0.607 | **0.624** | 0.614 | 0.483 | 0.679 | 0.561 | 0.576 | 0.600 | 0.586 | 0.473 | 0.673 | 0.553 |
| | ± 0.024 | ± 0.016 | ± 0.020 | ± 0.005 | ± 0.042 | ± 0.019 | ± 0.050 | ± 0.018 | ± 0.023 | ± 0.021 | ± 0.014 | ± 0.017 |
| VICReg | 0.706 | 0.592 | **0.661** | 0.615 | 0.665 | **0.635** | 0.674 | 0.591 | 0.641 | 0.619 | 0.649 | **0.631** |
| | ± 0.034 | ± 0.019 | ± 0.017 | ± 0.016 | ± 0.020 | ± 0.017 | ± 0.056 | ± 0.012 | ± 0.033 | ± 0.004 | ± 0.002 | ± 0.003 |
| Barlow Twins | **0.713** | 0.572 | 0.656 | 0.603 | 0.636 | 0.617 | **0.707** | 0.577 | **0.655** | 0.603 | 0.634 | 0.615 |
| | ± 0.014 | ± 0.008 | ± 0.005 | ± 0.010 | ± 0.064 | ± 0.027 | ± 0.004 | ± 0.003 | ± 0.002 | ± 0.020 | ± 0.011 | ± 0.008 |
| Concerto | 0.357 | 0.470 | 0.402 | 0.635 | 0.529 | 0.593 | — | — | — | — | — | — |
| | ± 0.000 | ± 0.029 | ± 0.012 | ± 0.016 | ± 0.028 | ± 0.021 | | | | | | |
| PCA | 0.656 | 0.320 | 0.522 | **0.651** | 0.348 | 0.530 | **0.656** | 0.320 | 0.522 | **0.651** | 0.348 | **0.530** |
| | ± 0.009 | ± 0.005 | ± 0.005 | ± 0.019 | ± 0.000 | ± 0.012 | ± 0.000 | ± 0.002 | ± 0.001 | ± 0.000 | ± 0.001 | ± 0.000 |

*Table H9.* Comparison of different multi-omics integration methods using the CLEAR pipeline. Data integration metrics were computed for the BMMC dataset.

| Method | Add | | | Concat | | | CLIP + Concat | | |
| | Bio | Batch | Total | Bio | Batch | Total | Bio | Batch | Total |
|---|---|---|---|---|---|---|---|---|---|
| SimCLR | 0.827 | 0.3 | 0.616 | 0.84 | 0.273 | 0.613 | 0.511 | **0.504** | 0.508 |
| | ± 0.078 | ± 0.057 | ± 0.05 | ± 0.093 | ± 0.058 | ± 0.065 | ± 0.223 | ± 0.094 | ± 0.166 |
| MoCo | **0.935** | 0.407 | **0.724** | 0.056 | **0.8** | 0.354 | 0.566 | 0.464 | 0.525 |
| | ± 0.07 | ± 0.019 | ± 0.045 | ± 0.065 | ± 0.000 | ± 0.039 | ± 0.157 | ± 0.161 | ± 0.049 |
| SimSiam | 0.453 | 0.174 | 0.341 | 0.506 | 0.21 | 0.387 | 0.197 | 0.364 | 0.264 |
| | ± 0.175 | ± 0.025 | ± 0.107 | ± 0.146 | ± 0.041 | ± 0.083 | ± 0.177 | ± 0.051 | ± 0.011 |
| NNCLR | 0.679 | 0.225 | 0.498 | 0.768 | 0.231 | 0.553 | 0.584 | 0.5 | 0.551 |
| | ± 0.171 | ± 0.041 | ± 0.113 | ± 0.105 | ± 0.034 | ± 0.06 | ± 0.088 | ± 0.079 | ± 0.066 |
| BYOL | 0.117 | **0.8** | 0.39 | 0.527 | 0.673 | 0.586 | 0.46 | 0.403 | 0.437 |
| | ± 0.109 | ± 0.000 | ± 0.066 | ± 0.029 | ± 0.074 | ± 0.03 | ± 0.123 | ± 0.152 | ± 0.115 |
| VICReg | 0.791 | 0.449 | 0.654 | **0.887** | 0.484 | **0.726** | **0.72** | 0.38 | **0.584** |
| | ± 0.089 | ± 0.014 | ± 0.052 | ± 0.035 | ± 0.009 | ± 0.022 | ± 0.079 | ± 0.055 | ± 0.056 |
| Barlow Twins | 0.717 | 0.262 | 0.535 | 0.852 | 0.27 | 0.62 | 0.706 | 0.352 | 0.565 |
| | ± 0.055 | ± 0.026 | ± 0.039 | ± 0.096 | ± 0.012 | ± 0.059 | ± 0.115 | ± 0.085 | ± 0.055 |

*Table H10.* Augmentation Parameters for the CLEAR (Han et al., 2022) augmentations on the left. Results for the ablation of all augmentations on the right, including the CLAIRE (Yan et al., 2023) augmentation denoted as MNN, and our BBKNN augmentation. Results stem from our ablation detailed in Appendix C.

| Augmentation | CLEAR | | | Ablation Result | | |
|---|---|---|---|---|---|---|
| | $\alpha$ | $\sigma$ | *knn* | $\alpha$ | $\sigma$ | *knn* |
| Masking | 0.2 | — | — | 0.5 | — | — |
| Gaussian Noise | 0.8 | 0.2 | — | 0.3 | 0.2 | — |
| InnerSwap | 0.1 | — | — | 0.3 | — | — |
| CrossOver | 0.25 | — | — | 0.1 | — | — |
| BBKNN | — | — | — | 0.9 | — | 3 |
| MNN | — | — | — | 0.5 | — | 3 |

*Table H11.* Batch correction benchmark for the Tabula Sapiens dataset (1.1 million cells) trained using the CLEAR pipeline (only 1 run) to show the scalability of our benchmark. The previously drawn conclusion that baselines outperform SSL methods in uni-modal batch correction holds. Generic SSL methods are good at batch correction but not preservation of the true biological variance.

| Method | Tabula Sapiens | | |
|---|---|---|---|
| | Bio | Batch | Total |
| SimCLR | 0.374 | 0.567 | 0.451 |
| MoCo | 0.342 | 0.541 | 0.421 |
| SimSiam | 0.237 | 0.430 | 0.314 |
| NNCLR | 0.312 | 0.454 | 0.368 |
| BYOL | 0.085 | **0.800** | 0.371 |
| VICReg | 0.327 | 0.657 | 0.459 |
| BarlowTwins | 0.376 | 0.427 | 0.396 |
| scVI | **0.723** | 0.254 | **0.536** |
| PCA | 0.538 | 0.297 | 0.442 |

