# OpenReview forum: "scSSL-Bench: Benchmarking Self-Supervised Learning for Single-Cell Data"
_ICML.cc/2025/Conference — ICML 2025 spotlightposter_

### Official Review · Reviewer_m4qB · 2025-03-02

**Overall Recommendation:** 3

**Summary:**

This paper proposed a benchmarking analysis for SSL method's application in single-cell data analysis.

## update after rebuttal

I raised my score.

**Claims And Evidence:**

Yes.

**Essential References Not Discussed:**

NA

**Experimental Designs Or Analyses:**

Yes. I have questions about this section, which is provided in my comments.

**Methods And Evaluation Criteria:**

Yes.

**Other Comments Or Suggestions:**

Please see my comments.

**Other Strengths And Weaknesses:**

NA.

**Questions For Authors:**

The authors proposed a benchmarking analysis for selecting the most suitable approach in performing self supervised learning (SSL) for single-cell data analysis. I have a couple of questions about the novelty and the meaning of this work. If the authors can address them, I may consider increasing the score. Please see my comments below

1. The motivation is not very clear to me. How to justify that the self supers learning approach is very powerful in analyzing single-cell data? Some cited methods such as scCLIP are not published. Could the authors provide more support to strengthen their position of self-supervised learning?

2. The baseline models should include more current research to support their conclusions. For example, there exists a couple of multi-omic data integration approaches which could be treated as potential better baselines (scGLUE, https://github.com/gao-lab/GLUE; scButterfly, https://www.nature.com/articles/s41467-024-47418-x; Monae, https://www.nature.com/articles/s41467-024-53355-6).

3. The selected tasks are also not interesting, especially cell-type annotation and modality prediction, which are actually two supervised learning tasks. We can not directly predict expression profiles or shall have levels without references. If the others can demonstrate the emergent abilities of SSL-based approaches in handling this task, I think it would be very interesting. Otherwise, the authors may focus on more SSL-related tasks, for example, only SSL-based methods can address and other baselines cannot perform, but it will be superbly hard, as simple tasks such as clustering are also well-explored.

4. Could the authors distinguish their contributions and the contributions of this paper (https://www.nature.com/articles/s42256-024-00934-3)? They are all talking about SSL for single-cell data analysis and the conclusions are similar.

5. Figure G1 is also not clear to me. From my understanding, the authors are working on single-cell data. Therefore, why do they include other modalities are examples? I suggest others to revise the figure using single-cell data as an example

**Relation To Broader Scientific Literature:**

I think the readers in single-cell analysis will be interested in reading this paper, but only limited to these people.

**Theoretical Claims:**

No, they do not have theoretical claims in the manuscript.

---

> ### Author Rebuttal · Authors · 2025-04-01
>
> We express sincere gratitude to the reviewer for providing feedback and raising several points about the validity of the work, which we address below and extend our evaluation accordingly. We hope that the reviewer will consider updating their review score if they find our comments and new results satisfying.
>
> **Motivation:** SSL is recognized for effectively learning robust representations from unlabeled data, enhancing downstream tasks. Many recent studies show the substantial advantages of SSL in single-cell analyses. Richter et al. (www.nature.com/articles/s42256-024-00934-3) demonstrates SSL methods excel in transfer learning, zero-shot, and cross-modality predictions, outperforming supervised approaches. The widely cited methods scVI , totalVI, CLEAR, CLAIRE, and Concerto show SSL's ability to capture biological heterogeneity and manage batch effects, better than supervised methods.
>
> **Additional Baselines:** Based on the reviewers’ suggestions, we added scButterfly and scTEL (www.nature.com/articles/s41540-024-00484-9) for multi-omics. The suggestions significantly improved the benchmark. scButterfly excels in cell typing, but scTEL underperforms in all tasks. However, our findings still indicate that SSL methods outperform other for multi-omics. New results in https://anonymous.4open.science/r/scSSL-Bench-Rebuttal/table_1.pdf, https://anonymous.4open.science/r/scSSL-Bench-Rebuttal/table_H4.pdf, https://anonymous.4open.science/r/scSSL-Bench-Rebuttal/table_H5.pdf. For uni-modal datasets, we added scGPT, scBERT and SCDC (https://ieeexplore.ieee.org/document/10086540), see our responses to X2so and KvUM.
>
> To clarify, our benchmark focuses on RNA-seq and CITE-seq data (RNA+protein) and is not directly comparable to scGLUE and Monae, which use RNA+ATAC. ATAC is a completely different modality that uses different methods. scGlue/Monae’s documentation covers only RNA+ATAC, without CITE-seq guidance.
>
> **Selected Tasks:** The selected tasks are integral to ongoing single-cell challenges (https://proceedings.mlr.press/v176/lance22a, https://openproblems.bio/results), supported by existing literature (scVI, totalVI, CLEAR, CLAIRE, Concerto), and relevant to a broader community (https://genomebiology.biomedcentral.com/articles/10.1186/s13059-020-1926-6).
> - *Batch Correction* is crucial for identifying true biological variation from experiment variability, e.g., platforms constantly change chemistry settings. Often, one cannot access a previous technology version and instead performs batch correction to unmask the biological signal, see prior studies www.nature.com/articles/s41592-018-0254-1, https://doi.org/10.1093/bioinformatics/btz625, www.nature.com/articles/s41592-021-01336-8, https://pubmed.ncbi.nlm.nih.gov/34062119.
> - *Cell Typing* maps new cells onto existing reference atlases and is vital for biological discoveries (www.nature.com/articles/s41587-021-01001-7, Concerto, scButterfly). Although the reviewer identifies cell typing as supervised, in the SSL context it is about unsupervised representation learning followed by minimal supervised inference.
> - *Missing Modality Prediction* infers unseen modalities, tests SSL method's ability to generalize across data types,  and enhances the utility of existing single-modality datasets. The inferred modalities can be used for the improved analysis (https://pubmed.ncbi.nlm.nih.gov/34062119, Concerto, scButterfly).
>
> **Contributions:** We cited Richter et al.from December 2024 in our initial paper and described the distinctions on line 36-55, col.2. To clarify our contributions and differences:
>
> - *Scope:* Richter et al. evaluate masked autoencoders (MAE), BYOL, and Barlow Twins on transfer learning, zero-shot, and fine-tuned SSL scenarios. Our scSSL-Bench assesses a broader range of several SSL methods, specialized and generic.
> - *Tasks:* Richter et al. focus on transfer learning. We evaluate batch correction, cell typing, and modality prediction for single- and multi-omics, addressing practical challenges from the community.
> - *Hyperparameters/Augmentations:* We address the impact of hyperparameters and augmentations for single-cell data, contributing valuable insights into optimal SSL configurations.
> - *Findings:* Richter et al. conclude in favor of MAE over contrastive methods at scale. We highlight different SSL methods perform optimally for different tasks and modalities, noting specialized SSL methods (scVI, CLAIRE) and scFMs excel at uni-modal batch correction, while generic SSL methods (VICReg, SimCLR) dominate for multi-modal data.
>
> **Figure G1** is an overview of benchmarked models showing structural differences. The visualized models can be applied to any input type. As we do not reference modalities in this figure, we're uncertain about the specific concern. We welcome suggestions to improve clarity.
>
> We appreciate the time you've taken to review our manuscript and provide your comments. We welcome any additional questions you might have.

---

> > ### Comment · Reviewer_m4qB · 2025-04-01
> >
> > Thank you for answering my questions, I raised my scores as weak acceptance as I still think the conclusions are not very interesting and it overlaps a lot with the previous SSL-based method for single-cell data analysis. But this work is very solid so I still want to vote for acceptance.

---

### Official Review · Reviewer_KvUM · 2025-03-10

**Overall Recommendation:** 4

**Summary:**

This paper proposes a self-supervised learning (SSL) benchmark for single-cell data. The authors tried twelve representative SSL methods and conducted comprehensive evaluations on eight datasets across three downstream tasks. The experimental designs are technically sound, and the paper is well-organized and written.

**Claims And Evidence:**

This work benchmarks the performance of representative SSL methods on single-cell data. The comparisons and ablation studies are well-designed, and the results could be a valuable reference for researchers interested in this area.

**Essential References Not Discussed:**

The current benchmarked methods are mostly discriminative ones. Currently, there are some generative single-cell SSL methods such as scBert (scBERT as a large-scale pretrained deep language model for cell type annotation of single-cell RNA-seq data, Nature Machine Intelligence 2022) and Geneformer (Transfer learning enables predictions in network biology, Nature 2023). The authors need to include these generative SSL methods in the benchmark as well.

Besides, the authors are also encouraged to include recent batch correction and data integration methods for single-cell data, such as Single-Cell RNA-Seq Debiased Clustering via Batch Effect Disentanglement (TNNLS 2024), scBridge embraces cell heterogeneity in single-cell RNA-seq and ATAC-seq data integration (Nature Communications 2023), etc.

**Experimental Designs Or Analyses:**

The chosen SSL methods and downstream tasks for benchmarking are representative. The experiments are well-designed to reflect the performance of different SSL methods and the effectiveness of different components in method design.

**Methods And Evaluation Criteria:**

The authors evaluated twelve representative SSL methods on eight datasets across three downstream tasks, including batch correction, cell type annotation, and missing modality prediction. The evaluations are comprehensive enough to help understand the effectiveness of different SSL methods on single-cell data. Ablation studies are also conducted to help interpret the importance of each component in the SSL methods.

**Other Comments Or Suggestions:**

When evaluating different representation dimensions, more choices like 128 and 256 are expected.

**Other Strengths And Weaknesses:**

The current datasets used for evaluation are all relatively small. It is interesting to see whether the same conclusions could be arrived at on much larger datasets such as the Human Fetal Atlas and Mouse Atlas.

When discussing the experimental results, the authors are encouraged to provide more in-depth explanations instead of plain descriptions. For example, in line 300, the authors write, ''these SSL methods prioritize batch correction over bio conservation as indicated by their high batch and low bio score.'' But why? More explanations are expected.

**Questions For Authors:**

I expect the authors to respond to my previous concerns. In addition, for missing modality prediction, the authors use the average of the nearest neighbors in the observed modality as the prediction results. However, the characteristics and neighbors in the two modalities could usually differ. In this case, is such a prediction paradigm reasonable?

**Relation To Broader Scientific Literature:**

This work could be a helpful reference for researchers interested in single-cell representation learning.

**Theoretical Claims:**

This work has no theoretical claims.

---

> ### Author Rebuttal · Authors · 2025-04-01
>
> We thank the reviewer for their constructive feedback. We appreciate your recognition that our paper could be a helpful reference for researchers interested in single-cell representation learning and that we conducted comprehensive evaluations. We address your questions and suggestions, which we have integrated into the updated manuscript to further improve the paper.
>
> **Additional Methods:** Following your’s and reviewer X2so’s suggestion, we extended the benchmark to also include single-cell Foundation Models (scFMs). As requested, we included scBERT for cell type annotation of single-cell RNA-seq data into our benchmark. In addition, we added Geneformer and scGPT for batch integration and cell type annotation, see https://anonymous.4open.science/r/scSSL-Bench-Rebuttal/table_1.pdf, https://anonymous.4open.science/r/scSSL-Bench-Rebuttal/table_H2.pdf and https://anonymous.4open.science/r/scSSL-Bench-Rebuttal/table_H3.pdf.
>
> We added scBridges to the introduction. Although scSSL-Bench currently leverages only RNA-seq and CITE-seq data, we cite scBridge and think it is valuable to extend the benchmark with ATAC-seq integration later. Moreover, we included results for SCDC (https://ieeexplore.ieee.org/document/10086540) for uni-modal tasks, and scButterfly and scTEL for multi-omics tasks. We accordingly updated the tables, see details in https://anonymous.4open.science/r/scSSL-Bench-Rebuttal.
>
> **Dataset Scale and Diversity**: We agree that incorporating very large datasets like the Human Fetal Atlas (\~4 million cells) and Mouse Atlas (\~300 thousand cells) would further validate our benchmark's scalability. While time constraints prevented their inclusion in this revision, we explicitly note this as a direction for future work in the manuscript.
>
> To demonstrate scalability, we have evaluated SSL methods and baselines on Tabula Sapiens (\~1.1 million cells, https://cellxgene.cziscience.com/collections/e5f58829-1a66-40b5-a624-9046778e74f5), see new table https://anonymous.4open.science/r/scSSL-Bench-Rebuttal/sapiens_bc.pdf. We'd also like to clarify that many of our existing datasets are substantial in size and widely used in the community, including Immune Cell Atlas (Conde et al., \~330,000 cells), multi-modal PBMC (Hao et al., \~160,000 cells), and BMMC (Lucken et al., \~90,000 cells), see https://anonymous.4open.science/r/scSSL-Bench-Rebuttal/datasets.pdf.
>
> **Results Discussion:** In the updated paper, we explain, wherever possible, the trade-offs between batch correction and bio conservation. Specifically, when describing how certain SSL methods prioritize batch correction over bio conservation, we discussed potential reasons, such as the nature of the SSL methods, the impact of specific augmentations, and the methods' underlying loss functions. For instance, in line 300, batch correction is prioritized since PBMC and ImunneCellAtlas datasets contain “harder” batch effects, SSL models try to correct the batch effects, neglecting bio conservation.
>
> **Representation Dimensions:** Our original rationale for selecting relatively small embeddings was guided by prior work in the literature, e.g., scVI www.nature.com/articles/s41592-018-0229-2, that demonstrated effective performance with lower-dimensional embeddings for single-cell analyses. We have extended the evaluation up to 1024 and come to the same conclusion. The representation dimensions of 64 or 128 reach a similar performance as 1024 while requiring less training time and memory for training and downstream tasks. We have updated Figure G4 with the following plots https://anonymous.4open.science/r/scSSL-Bench-Rebuttal/projection_HIC.pdf and https://anonymous.4open.science/r/scSSL-Bench-Rebuttal/projection_MCA.pdf.
>
> **Averaging for Missing Modality Prediction**: We agree that the characteristics and neighborhood structures of two modalities may differ, potentially affecting prediction accuracy. However, our choice of averaging nearest neighbors follows a common practice in the literature, e.g., Concerto www.nature.com/articles/s42256-022-00518-z.  Moreover, after adding scButterfly as suggested by reviewer m4qB that can generate missing modalities, we get a slightly better Person correlation with the nearest neighbor averaging using scButterfly embeddings (0.856) than generating proteins directly using scButterfly (0.84), see https://anonymous.4open.science/r/scSSL-Bench-Rebuttal/table_H5.pdf. We acknowledge the concern regarding potential discrepancies between modalities and now add this point to the discussion, highlighting that exploring more sophisticated prediction paradigms is an interesting avenue for future research.
>
> We sincerely thank you for your thoughtful review and constructive suggestions, which have significantly improved the quality and clarity of our paper. We welcome any additional questions that the reviewer might have.

---

> > ### Comment · Reviewer_KvUM · 2025-04-02
> >
> > I sincerely appreciate the effort the authors made in the rebuttal. My concerns have been well addressed and I would like to raise my score to accept.

---

### Official Review · Reviewer_X2so · 2025-03-11

**Overall Recommendation:** 5

**Summary:**

The authors present scSSL-Bench, a single-cell data benchmark that integrates 12 different approaches and 8 different datasets. The authors run extensive experimentations to answer three critical questions and provide invaluable insights and takeaways - This is no easy feat considering there are many moving parts due to different data augmentation, normalization, and training strategies.

**Claims And Evidence:**

Yes the authors pose 3 research questions which are extensively supported by detailed experimentations.

**Essential References Not Discussed:**

While I like how the authors extensively cited existing literature, I find the lack of any references to single-cell foundation models somewhat puzzling. Yes, there are lots of hype around single-cell FM, but still, I think a reader who doesn't understand nuances might be confused by lack of single-cell FM.

**Experimental Designs Or Analyses:**

I checked the experimental designs and analyses - Couldn't have been phrased better.

**Methods And Evaluation Criteria:**

Yes, it does.

**Other Comments Or Suggestions:**

- The dimension ablation seem intriguing, I am more used to much higher latent dimension (e.g., 512 in scGPT). So what is the rationale for only staying in the low regime (8,16,32,...)?

**Other Strengths And Weaknesses:**

I really appreciate the gargantuan effort that the authors had put into curating the benchmark. Having carried out effort myself, I know it's no easy feat, trying to draw overall takeaways/insights with so many moving parts. I think this will be a great contribution to the field where anyone not experienced enough in single-cell field can use it to jumpstart their research. I haven't tried out the github myself, but hopefully it is very straightforward to use for any newcomers. I hope the authors keep contributing to this benchmark, so that it can stand test of time.

Despite its strengths, I have few minor comments that are holding me back in giving higher scores.
- I think the authors needs to provide more detail on the 8 datasets used, e.g., sequencing technologies, how large is the gene panel for each dataset, since the readers would want to pick up the detail right away.
- My understanding is that there are way more publicly-available single-cell dataset than these 8. Why were these 8 chosen? Are there future plans to add much more datasets to this?
- As mentioned above, I find the lack of any discussion on single-cell FM quite puzzling. Yes, there is lots of hype, but I think they need to be either mentioned or benchmarked, since this will eventually be asked by end-users.
- Since augmentation in training is really crucial and important, ideally authors should provide one or two other SSL methods with augmentation ablations (only VICReg shown in the paper so far), to demonstrate the trend holds.

**Questions For Authors:**

See above

**Relation To Broader Scientific Literature:**

I think this is a timely addition that reflects the current single-cell literature mostly.

**Theoretical Claims:**

No theoretical claims were made in the paper.

---

> ### Author Rebuttal · Authors · 2025-04-01
>
> We thank the reviewer for their constructive feedback and highlighting the relevance of our benchmark and the quality of our experiments. In the following, we address your questions and suggestions, which have improved the quality of the paper and the benchmark.
>
> **Single-cell FM:** We acknowledge that Foundation Models (scFMs) have recently gained significant attention in the single-cell genomics community. Although scSSL-Bench evaluates a diverse set of SSL approaches, we added scGPT, Geneformer, and scBERT to the comparison, see https://anonymous.4open.science/r/scSSL-Bench-Rebuttal/table_1.pdf and https://anonymous.4open.science/r/scSSL-Bench-Rebuttal/table_H3.pdf.
>
> As suggested, we will concisely discuss and cite the scFMs in the Introduction to provide the necessary context for readers and will update the discussion according to our new results, which showcase that scFMs demonstrate strong performance for bio conservation in batch integration and good performance in cell typing. Our analysis reveals a substantial performance improvement in scGPT after fine-tuning compared to its zero-shot performance, underscoring the importance of fine-tuning scFMs. We will also add a section to the appendix detailing hyperparameters and fine-tuning.
>
> We also compare to one more single-modality method SCDC (https://ieeexplore.ieee.org/document/10086540) and two more multi-modal methods, scTEL (www.nature.com/articles/s41540-024-00484-9) and scButterfly (www.nature.com/articles/s41467-024-47418-x), as suggested by Reviewer m4qB, see https://anonymous.4open.science/r/scSSL-Bench-Rebuttal for new results.
>
> **Datasets Overview**: We have extended the dataset overview provided in Appendix B to include the sequencing technologies for each dataset, the number of features (genes, proteins), and the number of cells of a specific cell type, see details in https://anonymous.4open.science/r/scSSL-Bench-Rebuttal/datasets.pdf. For space consideration, we could not add this additional information to the main text.
>
> **Why These Datasets:** The selected datasets represent commonly used and established benchmarks in the single-cell literature, enabling direct comparisons with previous studies. For instance, these datasets have been employed in widely referenced studies and benchmarkings (e.g., www.nature.com/articles/s42256-024-00934-3, www.nature.com/articles/s42256-022-00518-z, www.nature.com/articles/s41540-024-00484-9, www.nature.com/articles/s41467-024-47418-x, https://academic.oup.com/bioinformatics/article/39/3/btad099/7055295, https://academic.oup.com/bib/article/23/5/bbac377/6695268), providing comparability across existing work. We now highlight this aspect in the manuscript. Furthermore, scSSL-Bench is engineered for scalability and easy extension. Our implementation with Hydra, a configuration management framework that enables flexible experiment configuration, parameter sweeping, and support for HPC environments, makes adding new datasets straightforward with minimal adjustments. This allows researchers to extend the benchmark with additional datasets. To this point, we also added the Tabula Sapiens dataset with 1.1 million cells to our evaluation, see https://anonymous.4open.science/r/scSSL-Bench-Rebuttal/sapiens_bc.pdf.
>
> **Augmentation Analysis**: Initially, we demonstrated augmentation ablations using VICReg due to its consistently strong performance. Following the reviewer's recommendation, we have extended our augmentation ablations to include SimCLR https://anonymous.4open.science/r/scSSL-Bench-Rebuttal/augmentations_simclr.pdf and MoCo https://anonymous.4open.science/r/scSSL-Bench-Rebuttal/augmentations_moco.pdf. This additional analysis confirms our original finding that masking is the most effective augmentation technique across all three SSL methods. Additionally, CrossOver shows competitive performance, especially for SimCLR.
>
> **Representation Dimensions:** Our original rationale for selecting relatively small embeddings was guided by prior work in the literature, e.g., scVI www.nature.com/articles/s41592-018-0229-2, that demonstrated effective performance with lower-dimensional embeddings for single-cell analyses. We have extended the evaluation up to 1024 and come to the same conclusion. The representation dimensions of 64 or 128 reach a similar performance as 1024 while requiring less training time and memory for training and downstream tasks. We have updated Figure G4 with the following plots https://anonymous.4open.science/r/scSSL-Bench-Rebuttal/projection_HIC.pdf and https://anonymous.4open.science/r/scSSL-Bench-Rebuttal/projection_MCA.pdf.
>
> We appreciate the reviewer's thoughtful feedback and have made substantial improvements to the manuscript accordingly. These changes have significantly strengthened our work and we hope scSSL-Bench will serve as a valuable resource for the single-cell and machine learning communities. We are happy to answer any additional questions that the reviewers might have.

---

> > ### Comment · Reviewer_X2so · 2025-04-02
> >
> > All my questions were throughly answered (more than sufficient) and I have adjusted my score accordingly - I think it's a really good contribution to the community.

---

### Decision · Program_Chairs · 2025-05-01

**Decision:**

Accept (spotlight poster)

**Comment:**

This paper presents scSSL-Bench, a comprehensive and timely benchmarking platform that evaluates 12 SSL methods across 8 single-cell datasets and three downstream tasks. The authors offer detailed methodological comparisons, ablation studies on data augmentations and embedding dimensions, and practical recommendations tailored for batch correction, cell-type annotation, and modality prediction. All reviewers ultimately supported acceptance, with two strong supports acknowledging the paper’s rigor despite concerns of overlap with existing work. Importantly, the authors thoughtfully and thoroughly addressed reviewer concerns, significantly enhancing the benchmark's scope and reproducibility. Given the clarity and comprehensiveness of this benchmark, I recommend acceptance.